# How Data Augmentation affects Optimization for Linear Regression

**Boris Hanin**[*]
Department of Operations Research
and Financial Engineering
Princeton University
bhanin@princeton.edu

**Yi Sun**[*]
Department of Statistics
University of Chicago
yisun@statistics.uchicago.edu

## Abstract

Though data augmentation has rapidly emerged as a key tool for optimization in modern machine learning, a clear picture of how augmentation schedules affect optimization and interact with optimization hyperparameters such as learning rate is nascent. In the spirit of classical convex optimization and recent work on implicit bias, the present work analyzes the effect of augmentation on optimization in the simple convex setting of linear regression with MSE loss.

We find joint schedules for learning rate and data augmentation scheme under which augmented gradient descent provably converges and characterize the resulting minimum. Our results apply to arbitrary augmentation schemes, revealing complex interactions between learning rates and augmentations even in the convex setting. Our approach interprets augmented (S)GD as a stochastic optimization method for a time-varying sequence of proxy losses. This gives a unified way to analyze learning rate, batch size, and augmentations ranging from additive noise to random projections. From this perspective, our results, which also give rates of convergence, can be viewed as Monro-Robbins type conditions for augmented (S)GD.

## 1 Introduction

Data augmentation, a popular set of techniques in which data is augmented (i.e. modified) at every optimization step, has become increasingly crucial to training models using gradient-based optimization. However, in modern overparametrized settings where there are many different minimizers of the training loss, the specific minimizer selected by training and the quality of the resulting model can be highly sensitive to choices of augmentation hyperparameters. As a result, practitioners use methods ranging from simple grid search to Bayesian optimization and reinforcement learning [8, 9, 17] to select and schedule augmentations by changing hyperparameters over the course of optimization. Such approaches, while effective, often require extensive compute and lack theoretical grounding.

These empirical practices stand in contrast to theoretical results from the implicit bias and stochastic optimization literature. The extensive recent literature on implicit bias [15, 29, 32] gives provable guarantees on which minimizer of the training loss is selected by GD and SGD in simple settings, but considers cases without complex scheduling. On the other hand, classical theorems in stochastic optimization, building on the Monro-Robbins theorem in [25], give provably optimal learning rate schedules for strongly convex objectives. However, neither line of work addresses the myriad augmentation and hyperparameter choices crucial to gradient-based training effective in practice.

The present work takes a step towards bridging this gap. We consider two main questions for a learning rate schedule and data augmentation policy:

---

[*]Equal contribution

35th Conference on Neural Information Processing Systems (NeurIPS 2021).

1. When and at what rate will optimization converge?

2. Assuming optimization converges, what point does it converge to?

To isolate the effect *on optimization* of jointly scheduling learning rate and data augmentation schemes, we consider these questions in the simple convex model of linear regression with MSE loss:

$$\mathcal{L}(W; \mathcal{D}) = \frac{1}{N} \|Y - WX\|_F^2. \tag{1.1}$$

In this setting, we analyze the effect of the data augmentation policy on the optimization trajectory $W_t$ of augmented (stochastic) gradient descent[2], which follows the update equation

$$W_{t+1} = W_t - \eta_t \nabla_W \mathcal{L}(W; \mathcal{D}_t)\big|_{W=W_t}. \tag{1.2}$$

Here, the dataset $\mathcal{D} = (X, Y)$ contains $N$ datapoints arranged into data matrices $X \in \mathbb{R}^{n \times N}$ and $Y \in \mathbb{R}^{p \times N}$ whose columns consist of inputs $x_i \in \mathbb{R}^n$ and outputs $y_i \in \mathbb{R}^p$. In this context, we take a flexible definition *data augmentation scheme* as any procedure that consists, at every step of optimization, of replacing the dataset $\mathcal{D}$ by a randomly augmented variant which we denote by $\mathcal{D}_t = (X_t, Y_t)$. This framework is flexible enough to handle SGD and commonly used augmentations such as additive noise [14], CutOut [12], SpecAugment [23], Mixup [35], and label-preserving transformations (e.g. color jitter, geometric transformations [26])).

We give a general answer to Questions 1 and 2 for arbitrary data augmentation schemes. Our main result (Theorem 3.1) gives sufficient conditions for optimization to converge in terms of the learning rate schedule and simple $2^{\text{nd}}$ and $4^{\text{th}}$ order moment statistics of augmented data matrices. When convergence occurs, we explicitly characterize the resulting optimum in terms of these statistics. We then specialize our results to (S)GD with modern augmentations such as additive noise [14] and random projections (e.g. CutOut [12] and SpecAugment [23]). In these cases, we find learning rate and augmentation parameters which ensure convergence with rates to the minimum norm optimum for overparametrized linear regression. To sum up, our main contributions are:

1. We analyze *arbitrary* data augmentation schemes for linear regression with MSE loss, obtaining explicit sufficient conditions on the joint schedule of the data augmentation policy and the learning rate for (stochastic) gradient descent that guarantee convergence with rates in Theorems 3.1 and 3.2. The resulting augmentation-dependent optimum encodes the ultimate effect of augmentation on optimization, and we characterize it in Theorem 3.1. Our results generalize Monro-Robbins theorems [25] to situations where the stochastic optimization objective may change at each step.

2. We specialize our results to (stochastic) gradient descent with additive input noise (§4) or random projections of the input (§5), a proxy for the popular CutOut and SpecAugment augmentations [12, 23]. In each case, we find that jointly scheduling learning rate and augmentation strength is critical for allowing convergence with rates to the minimum norm optimizer. We find specific schedule choices which guarantee this convergence with rates (Theorems 4.1, 4.2, and 5.1) and validate our results empirically (Figure 4.1). This suggests explicitly adding learning rate schedules to the search space for learned augmentations as in [8, 9], which we leave to future work.

## 2 Related Work

In addition to the extensive empirical work on data augmentation cited elsewhere in this article, we briefly catalog other theoretical work on data augmentation and learning rate schedules. The latter were first considered in the seminal work [25]. This spawned a vast literature on *rates* of convergence for GD, SGD, and their variants. We mention only the relatively recent articles [1, 11, 4, 27, 22] and the references therein. The last of these, namely [22], finds optimal choices of learning rate and batch size for SGD in the overparametrized linear setting.

A number of articles have also pointed out in various regimes that data augmentation and more general transformations such as feature dropout correspond in part to $\ell_2$-type regularization on model parameters, features, gradients, and Hessians. The first article of this kind of which we are aware is [3],

---

[2]Both GD and SGD fall into our framework. To implement SGD, we take $\mathcal{D}_t$ to be a subset of $\mathcal{D}$.

which treats the case of additive Gaussian noise (see §4). More recent work in this direction includes [5, 30, 19, 21]. There are also several articles investigating *optimal* choices of $\ell_2$-regularization for linear models (cf e.g. [33, 31, 2]). These articles focus directly on the generalization effects of ridge-regularized minima but not on the dynamics of optimization. We also point the reader to [20], which considers optimal choices for the weight decay coefficient empirically in neural networks and analytically in simple models.

We also refer the reader to a number of recent attempts to characterize the benefits of data augmentation. In [24], for example, the authors quantify how much augmented data, produced via additive noise, is needed to learn positive margin classifiers. [6], in contrast, focuses on the case of data invariant under the action of a group. Using the group action to generate label-preserving augmentations, the authors prove that the variance of any function depending only on the trained model will decrease. This applies in particular to estimators for the trainable parameters themselves. [10] shows augmented $k$-NN classification reduces to a kernel method for augmentations transforming each datapoint to a finite orbit of possibilities. It also gives a second order expansion for the proxy loss of a kernel method under such augmentations and interprets how each term affects generalization. Finally, the article [34] considers both label preserving and noising augmentations, pointing out the conceptually distinct roles such augmentations play.

## 3 Time-varying Monro-Robbins for linear models under augmentation

We seek to isolate the impact of data augmentation on optimization in the simple setting of augmented (stochastic) gradient descent for linear regression with the MSE loss (1.1). Since the augmented dataset $\mathcal{D}_t$ at time $t$ is a stochastic function of $\mathcal{D}$, we may view the update rule (1.2) as a form of stochastic optimization for the *proxy loss at time $t$*

$$\overline{\mathcal{L}}_t(W) := \mathbb{E}_{\mathcal{D}_t}\left[\mathcal{L}(W; \mathcal{D}_t)\right] \tag{3.1}$$

which uses an unbiased estimate of the gradient of $\mathcal{L}(W; \mathcal{D}_t)$ from a single draw of $\mathcal{D}_t$. The connection between data augmentation and this proxy loss was introduced in [3, 5], but we now consider it in the context of stochastic optimization. In particular, we consider scheduling the augmentation, which allows the distribution of $\mathcal{D}_t$ to change with $t$ and thus enables optimization to converge to points which are not minimizers of the proxy loss $\overline{\mathcal{L}}_t(W)$ at any fixed time.

Our main results, Theorems 3.1 and 3.2, provide sufficient conditions for jointly scheduling learning rates and general augmentation schemes to guarantee convergence of augmented gradient descent in the linear regression model (1.1). Before stating them, we first give examples of augmentations falling into our framework, which we analyze using our general results in §4 and §5.

- **Additive Gaussian noise:** For SGD with batch size $B_t$ and noise level $\sigma_t > 0$, this corresponds to $X_t = c_t(XA_t + \sigma_t \cdot G_t)$ and $Y_t = c_t Y A_t$, where $G_t$ is a matrix of i.i.d. standard Gaussians, $A_t \in \mathbb{R}^{N \times B_t}$ has i.i.d. columns with a single non-zero entry equal to 1 chosen uniformly at random and $c_t = \sqrt{N/B_t}$ is a normalizing factor. The proxy loss is

$$\overline{\mathcal{L}}_t(W) = \mathcal{L}(W; \mathcal{D}) + \sigma_t^2 \|W\|_F^2, \tag{3.2}$$

  which adds an $\ell_2$ penalty. We analyze this case in §4.

- **Random projection:** This corresponds to $X_t = \Pi_t X$ and $Y_t = Y$, where $\Pi_t$ is an orthogonal projection onto a random subspace. For $\gamma_t = \text{tr}(\Pi_t)/n$, the proxy loss is

$$\overline{\mathcal{L}}_t(W) = \frac{1}{N}\|Y - \gamma_t W X\|_F^2 + \frac{1}{N}\gamma_t(1 - \gamma_t)\frac{1}{n}\|X\|_F^2 \cdot \|W\|_F^2 + O(n^{-1}),$$

  adding a data-dependent $\ell_2$ penalty and applying Stein-type shrinkage on input data. We analyze this in §5.

In addition to these augmentations, the augmentations below also fit into our framework, and Theorems 3.1 and 3.2 apply. However, in these cases, explicitly characterizing the learned minimum beyond the general description given in Theorems 3.1 and 3.2 is more difficult, and we thus leave interpretation of these specializations to future work.

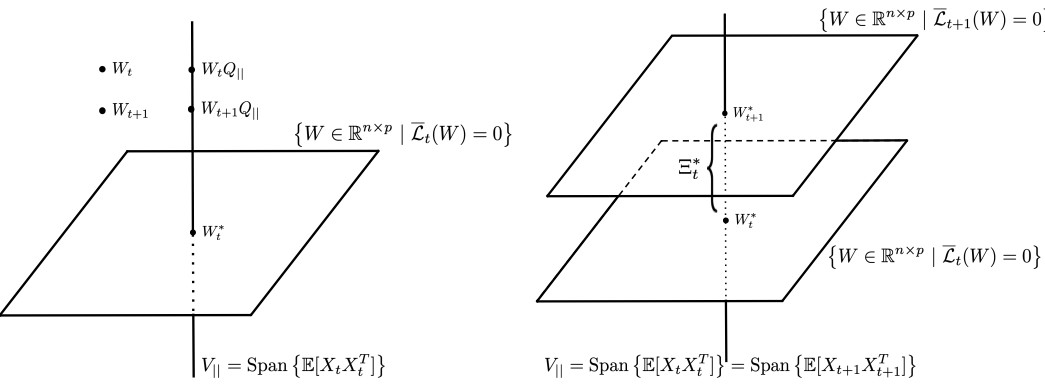

(a) The time $t$ proxy loss $\overline{\mathcal{L}}_t$ is non-degenerate on $V_{\parallel}$ with minimal norm minimizer $W_t^*$. The increment $W_{t+1} - W_t$ is in $V_{\parallel}$, so only the projection $W_t Q_{\parallel}$ of $W_t$ onto $V_{\parallel}$ changes.

(b) The proxy losses $\overline{\mathcal{L}}_t$ and $\overline{\mathcal{L}}_{t+1}$ at consecutive times will generally have different minimal norm minimizers $W_t^*$, $W_{t+1}^*$. The increment $\Xi_t^* = W_{t+1}^* - W_t^*$ measures this change.

Figure 3.1: Schematic diagrams of augmented optimization in the parameter space $\mathbb{R}^{n \times p}$.

- **Label-preserving transformations:** For a 2-D image viewed as a vector $x \in \mathbb{R}^n$, geometric transforms (with pixel interpolation) or other label-preserving transforms such as color jitter take the form of linear transforms $\mathbb{R}^n \to \mathbb{R}^n$. We may implement such augmentations in our framework by $X_t = A_t X$ and $Y_t = Y$ for some random transform matrix $A_t$.

- **Mixup:** To implement Mixup, we can take $X_t = X A_t$ and $Y_t = Y A_t$, where $A_t \in \mathbb{R}^{N \times B_t}$ has i.i.d. columns containing two random non-zero entries equal to $1 - c_t$ and $c_t$ with mixing coefficient $c_t$ drawn independently from a Beta$(\alpha_t, \alpha_t)$ distribution for a parameter $\alpha_t$.

## 3.1 A general time-varying Monro-Robbins theorem

Given an augmentation scheme for the overparameterized linear model (1.1), the time $t$ gradient update at learning rate $\eta_t$ is

$$W_{t+1} := W_t + \frac{2\eta_t}{N} \cdot (Y_t - W_t X_t) X_t^{\mathsf{T}}, \tag{3.3}$$

where $\mathcal{D}_t = (X_t, Y_t)$ is the augmented dataset at time $t$. The minimum norm minimizer of the corresponding proxy loss $\overline{\mathcal{L}}_t$ (see (3.1)) is

$$W_t^* := \mathbb{E}[Y_t X_t^{\mathsf{T}}] \mathbb{E}[X_t X_t^{\mathsf{T}}]^+, \tag{3.4}$$

where $\mathbb{E}[X_t X_t^{\mathsf{T}}]^+$ denotes the Moore-Penrose pseudo-inverse (see Figure 3.1a). In this section we state a rigorous result, Theorem 3.1, giving sufficient conditions on the learning rate $\eta_t$ and distributions of the augmented matrices $X_t, Y_t$ under which augmented gradient descent converges. To state it, note that (3.3) implies that each row of $W_{t+1} - W_t$ is contained in the column span of the Hessian $X_t X_t^{\mathsf{T}}$ of the augmented loss and therefore almost surely belongs to the subspace

$$V_{\parallel} := \text{column span of } \mathbb{E}[X_t X_t^{\mathsf{T}}] \subseteq \mathbb{R}^n, \tag{3.5}$$

as illustrated in Figure 3.1a. The reason is that, in the orthogonal complement to $V_{\parallel}$, the augmented loss $\mathcal{L}(W; \mathcal{D}_t)$ has zero gradient with probability 1. To ease notation, we assume that $V_{\parallel}$ is independent of $t$. This assumption holds for additive Gaussian noise, random projection, MixUp, SGD, and their combinations. It is not necessary in general, however, and we refer the interested reader to Remark B.2 in the Appendix for how to treat the general case.

Let us denote by $Q_{\parallel} : \mathbb{R}^n \to \mathbb{R}^n$ the orthogonal projection onto $V_{\parallel}$ (see Figure 3.1a). As we already pointed out, at time $t$, gradient descent leaves the matrix of projections $W_t(\text{Id} - Q_{\parallel})$ of each row of $W_t$ onto the orthogonal complement of $V_{\parallel}$ unchanged. In contrast, $\|W_t Q_{\parallel} - W_t^*\|_F$ decreases at a rate governed by the smallest positive eigenvalue $\lambda_{\min, V_{\parallel}}\left(\mathbb{E}\left[X_t X_t^{\mathsf{T}}\right]\right)$ of the Hessian for the proxy

loss $\overline{\mathcal{L}}_t$, which is obtained by restricting its full Hessian $\mathbb{E}\left[X_t X_t^\mathsf{T}\right]$ to $V_\|$. Moreover, whether and at what rate $W_t Q_\| - W_t^*$ converges to $0$ depends on how quickly the distance

$$\Xi_t^* := W_{t+1}^* - W_t^* \tag{3.6}$$

between proxy loss optima at successive steps tends to zero (see Figure 3.1b).

**Theorem 3.1** (Special case of Theorem B.1). *Suppose that $V_\|$ is independent of $t$, that the learning rate satisfies $\eta_t \to 0$, that the proxy optima satisfy*

$$\sum_{t=0}^\infty \|\Xi_t^*\|_F < \infty, \tag{3.7}$$

*ensuring the existence of a limit $W_\infty^* := \lim_{t\to\infty} W_t^*$, that*

$$\sum_{t=0}^\infty \eta_t \lambda_{min,V_\|}\left(\mathbb{E}[X_t X_t^\mathsf{T}]\right) = \infty \tag{3.8}$$

*and finally that*

$$\sum_{t=0}^\infty \eta_t^2 \mathbb{E}\left[\|X_t X_t^\mathsf{T} - \mathbb{E}[X_t X_t^\mathsf{T}]\|_F^2 + \|Y_t X_t^\mathsf{T} - \mathbb{E}[Y_t X_t^\mathsf{T}]\|_F^2\right] < \infty. \tag{3.9}$$

*Then, for any initialization $W_0$, we have that $W_t Q_\|$ converges in probability to $W_\infty^*$.*

If the same augmentation is applied with different strength parameters at each step $t$ (e.g. the noise level $\sigma_t^2$ for additive Gaussian noise), we may specialize Theorem 3.1 to this augmentation scheme. More precisely, translating conditions (3.7), (3.8), (3.9) into conditions on the learning rate and augmentation strength gives conditions on the schedule for $\eta_t$ and these strength parameters to ensure convergence. In §4 and §5, we do this for additive Gaussian noise and random projections.

When the augmentation scheme is static in $t$, Theorem 3.1 reduces to a standard Monro-Robbins theorem [25] for the (static) proxy loss $\overline{\mathcal{L}}_t(W)$. As in that setting, condition (3.8) enforces that the learning trajectory travels far enough to reach an optimum, and the summand in Condition (3.9) bounds the variance of the gradient of the augmented loss $\mathcal{L}(W; \mathcal{D}_t)$ to ensure the total variance of the stochastic gradients is summable. Condition (3.7) is new and enforces that the minimizers $W_t^*$ of the proxy losses $\overline{\mathcal{L}}_t(W)$ change slowly enough for augmented optimization procedure to keep pace.

## 3.2 Convergence rates and scheduling for data augmentation

Refining the proof of Theorem 3.1 gives rates of convergence for the projections $W_t Q_\|$ of the weights onto $V_\|$ to the limiting optimum $W_\infty^*$. When the quantities in Theorem 3.1 have power law decay, we obtain the following result.

**Theorem 3.2** (Special case of Theorem B.4). *Suppose $V_\|$ is independent of $t$ and the learning rate satisfies $\eta_t \to 0$. Moreover assume that for some $0 < \alpha < 1 < \beta_1, \beta_2$ and $\gamma > \alpha$ we have*

$$\eta_t \lambda_{min,V_\|}\left(\mathbb{E}[X_t X_t^\mathsf{T}]\right) = \Omega(t^{-\alpha}), \qquad \|\Xi_t^*\|_F = O(t^{-\beta_1}) \tag{3.10}$$

*as well as*

$$\eta_t^2 \mathbb{E}[\|X_t X_t^\mathsf{T} - \mathbb{E}[X_t X_t^\mathsf{T}]\|_2^2] = O(t^{-\gamma}) \tag{3.11}$$

*and*

$$\eta_t^2 \mathbb{E}\left[\|\mathbb{E}[W_t](X_t X_t^\mathsf{T} - \mathbb{E}[X_t X_t^\mathsf{T}]) - (Y_t X_t^\mathsf{T} - \mathbb{E}[Y_t X_t^\mathsf{T}])\|_F^2\right] = O(t^{-\beta_2}). \tag{3.12}$$

*Then, for any initialization $W_0$, we have for any $\varepsilon > 0$ the following convergence in probability:*

$$t^{\min\{\beta_1 - 1, \frac{\beta_2 - \alpha}{2}\} - \varepsilon} \|W_t Q_\| - W_\infty^*\|_F \xrightarrow{p} 0.$$

It may be surprising that $\mathbb{E}[W_t]$ appears in condition (3.12). Note that $\mathbb{E}[W_t]$ is the gradient descent trajectory for the time-varying sequence of deterministic proxy losses $\overline{\mathcal{L}}_t(W)$. To apply Theorem 3.2, one may first study this deterministic problem to show that $\mathbb{E}[W_t]$ converges to $W_\infty^*$ at some rate and then use (3.12) to obtain rates of convergence of the true stochastic trajectory $W_t$ to $W_\infty^*$.

In §4 and §5 below, we specialize Theorems 3.1 and 3.2 to obtain rates of convergence for specific augmentations. Optimizing the learning rate and augmentation parameter schedules in Theorem 3.2 yields power law schedules with convergence rate guarantees in these settings.

# 4 Special Case: Additive Gaussian Noise

We now specialize our main results Theorem 3.1 and 3.2 to the commonly used additive noise augmentation [14]. Under gradient descent, this corresponds to taking augmented data matrices

$$X_t = X + \sigma_t G_t \qquad \text{and} \qquad Y_t = Y,$$

where $G_t \in \mathbb{R}^{n \times N}$ are independent matrices with i.i.d. standard Gaussian entries, and $\sigma_t > 0$ is a strength parameter. Under SGD (with replacement), this corresponds to augmented data matrices

$$X_t = c_t(XA_t + \sigma_t \cdot G_t) \qquad \text{and} \qquad Y_t = c_t Y A_t,$$

where $A_t \in \mathbb{R}^{N \times B_t}$ has i.i.d. columns with a single non-zero entry equal to 1 chosen uniformly at random and $c_t = \sqrt{N/B_t}$ is a normalizing factor. In both cases, the proxy loss is

$$\mathcal{L}_{\sigma_t}(W) := \frac{1}{N}\|Y - WX\|_F^2 + \sigma_t^2\|W\|_F^2, \tag{4.1}$$

which to our knowledge was first discovered in [3].

Before stating our precise results, we first illustrate how jointly scheduling learning rate and augmentation strength impacts GD for overparameterized linear regression, where

$$N = \#\text{data points} < \text{input dimension} = n. \tag{4.2}$$

The inequality (4.2) ensures $\mathcal{L}(W; \mathcal{D})$ has infinitely many minima, of which we consider the *minimum norm* minimizer

$$W_{\min} := YX^\mathsf{T}(XX^\mathsf{T})^+$$

most desirable. Notice that steps of vanilla gradient descent

$$W_{t+1} = W_t - \frac{2\eta_t}{N} \cdot (Y - W_t X)X^\mathsf{T} \tag{4.3}$$

change the rows of the weight matrix $W_t$ only in the column space $V_\| = \text{colspan}(XX^\mathsf{T}) \subseteq \mathbb{R}^n$. Because $V_\| \neq \mathbb{R}^n$ by the overparameterization assumption (4.2), minimizing $\mathcal{L}(W; \mathcal{D})$ *without augmentation* cannot change $W_{t,\perp}$, the matrix whose rows are the components of the rows of $W_t$ orthogonal to $V_\|$. This means that GD converges to the minimal norm optimizer $W_{\min}$ only when each row of $W_0$ belongs to $V_\|$. As this explicit initialization may not be available for more general models, we seek to find augmentation schedules which allow GD or SGD to converge to $W_{\min}$ without it, in the spirit of recent studies on implicit bias of GD.

## 4.1 Joint schedules for augmented GD with additive noise to converge to $W_{\min}$

We specialize Theorems 3.1 and 3.2 to additive Gaussian noise to show that when the learning rate $\eta_t$ and noise strength $\sigma_t$ are jointly scheduled to converge to 0 at appropriate rates, augmented gradient descent can find the minimum norm optimizer $W_{\min}$.

**Theorem 4.1.** *Consider any joint schedule of the learning rate $\eta_t$ and noise variance $\sigma_t^2$ in which both $\eta_t$ and $\sigma_t^2$ tend to 0 and $\sigma_t$ is non-decreasing. If the joint schedule satisfies*

$$\sum_{t=0}^\infty \eta_t \sigma_t^2 = \infty \qquad \text{and} \qquad \sum_{t=0}^\infty \eta_t^2 \sigma_t^2 < \infty, \tag{4.4}$$

*then the weights $W_t$ converge in probability to the minimal norm optimum $W_{min}$ regardless of the initialization. Moreover, the first condition in (4.4) is necessary for $\mathbb{E}[W_t]$ to converge to $W_{\min}$.*

*If we further have $\eta_t = \Theta(t^{-x})$ and $\sigma_t^2 = \Theta(t^{-y})$ with $x, y > 0$, $x + y < 1$, and $2x + y > 1$ so that $\eta_t$ and $\sigma_t^2$ satisfy (4.4), then for small $\varepsilon > 0$, we have $t^{\min\{y, \frac{1}{2}x\} - \varepsilon}\|W_t - W_{min}\|_F \xrightarrow{p} 0$.*

The conditions of (4.4) require that $\eta_t$ and $\sigma_t$ be jointly scheduled correctly to ensure convergence to $W_{\min}$ and are akin to the Monro-Robbins conditions [25] for convergence of stochastic gradient methods. We now give an heuristic explanation for why the first condition from (4.4) is necessary. In this setting, the *average* trajectory of augmented gradient descent

$$\mathbb{E}[W_{t+1}] = \mathbb{E}[W_t] - \eta \nabla_W \mathcal{L}_{\sigma_t}(W)\big|_{W = \mathbb{E}[W_t]} \tag{4.5}$$

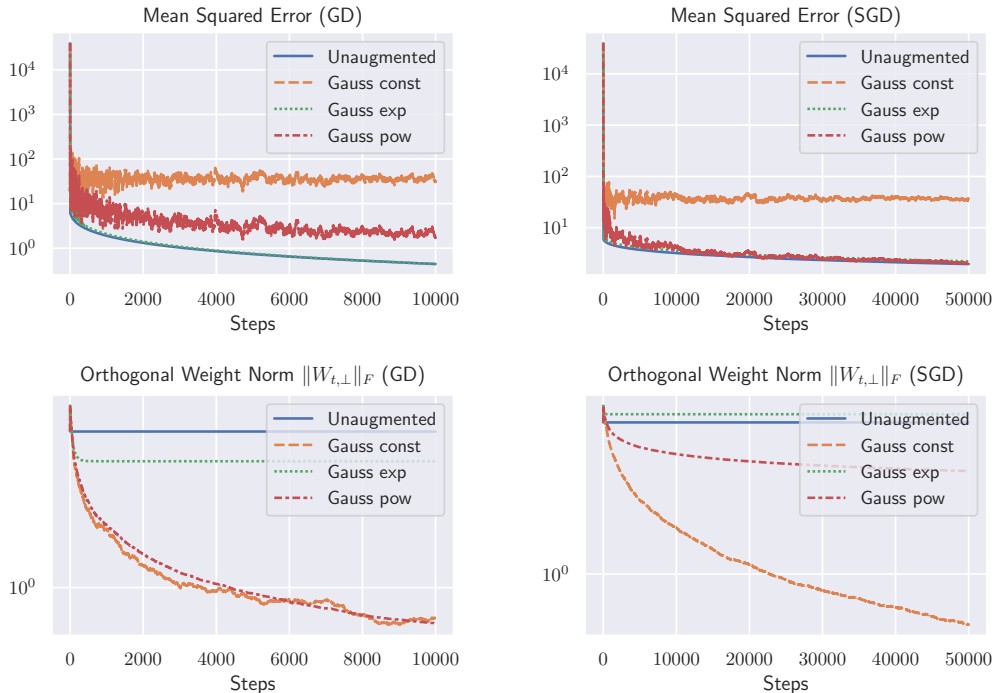

Figure 4.1: MSE and $\|W_{t,\perp}\|_F$ for optimization trajectories of GD with additive Gaussian noise augmentation and SGD with additive Gaussian noise augmentation under different augmentation schedules. For both GD and SGD, jointly scheduling learning rate and noise variance to have polynomial decay is necessary for optimization to converge to the minimal norm solution $W_{\min}$. Gauss const, Gauss exp, and Gauss pow have Gaussian noise augmentation with $\sigma_t^2 = 2, 2e^{-0.02t}, 2(1 + \frac{t}{50})^{-0.33}$, respectively. All other details are given in §4.3.

is given by gradient descent on the ridge regularized losses $\mathcal{L}_{\sigma_t}(W)$. If $\sigma_t \equiv \sigma > 0$ is constant, then $\mathbb{E}[W_t]$ will converge to the unique minimizer $W_\sigma^*$ of the ridge regularized loss $\mathcal{L}_\sigma$. This point $W_\sigma^*$ has zero orthogonal component, but *does not* minimize the original loss $\mathcal{L}$.

To instead minimize $\mathcal{L}$, we must choose a schedule satisfying $\sigma_t \to 0$. For the expected optimization trajectory to converge to $W_{\min}$ for such a schedule, the matrix $\mathbb{E}[W_{t,\perp}]$ of components of rows of $\mathbb{E}[W_t]$ orthogonal to $V_\parallel$ must converge to 0. The GD steps for this matrix yield

$$\mathbb{E}[W_{t+1,\perp}] = (1 - \eta_t \sigma_t^2)\mathbb{E}[W_{t,\perp}] = \prod_{s=0}^{t}(1 - \eta_s \sigma_s^2)\mathbb{E}[W_{0,\perp}]. \tag{4.6}$$

Because $\eta_t \sigma_t^2$ approaches 0, this implies the necessary condition $\sum_{t=0}^{\infty} \eta_t \sigma_t^2 = \infty$ for $\mathbb{E}[W_{t,\perp}] \to 0$.

This argument illustrates a key intuition behind the conditions (4.4). The augmentation strength $\sigma_t$ must decay to 0 to allow convergence to a true minimizer of the training loss, but this convergence must be carefully tuned to allow the implicit regularization of the noising augmentation to kill the orthogonal component of $W_t$ in expectation. In a similar manner, the second expression in (4.4) measures the total variance of the gradients and ensures that only a finite amount of noise is injected into the optimization.

Although Theorem 4.1 is stated for additive Gaussian noise, an analogous version holds for arbitrary additive noise with bounded moments. Moreover, optimizing over $x, y$, the fastest rate of convergence guaranteed by Theorem 4.1 is obtained by setting $\eta_t = t^{-2/3+\varepsilon}$, $\sigma_t^2 = t^{-1/3}$ and results in a $O(t^{-1/3+\varepsilon})$ rate of convergence. It is not evident that this is best possible, however.

### 4.2 Joint schedules for augmented SGD with additive noise to converge to $W_{\min}$

To conclude our discussion of additive noise augmentations, we present the following result on convergence to $W_{\min}$ in the presence of both Gaussian noise and SGD (where datapoints in each batch are selected with replacement).

**Theorem 4.2.** *Suppose $\sigma_t^2 \to 0$ is decreasing, $\eta_t \to 0$, and we have*

$$\sum_{t=0}^{\infty} \eta_t \sigma_t^2 = \infty \qquad and \qquad \sum_{t=0}^{\infty} \eta_t^2 < \infty. \tag{4.7}$$

*Then the trajectory $W_t$ of SGD with additive noise converges in probability to $W_{min}$ for any initialization. If we further have $\eta_t = \Theta(t^{-x})$ and $\sigma_t^2 = \Theta(t^{-y})$ with $x > \frac{1}{2}$, $y > 0$ and $x + y < 1$, then for any $\varepsilon > 0$ we have that $t^{\min\{y, \frac{1}{2}x\} - \varepsilon} \|W_t - W_{min}\|_F \xrightarrow{p} 0$.*

Theorem 4.2 is the analog of Theorem 4.1 for mini-batch SGD and provides an example where our framework can handle the *composition* of two augmentations, namely additive noise and mini-batching. The difference between conditions (4.7) for SGD and (4.4) for GD accounts for the fact that the batch selection changes the scale of the gradient variance at each step. Finally, Theorem 4.2 reveals a qualitative difference between SGD with and without additive noise. If $\eta_t$ has power law decay, the convergence of noiseless SGD (Theorem F.1) is exponential in $t$, while Theorem 4.2 gives power law rates.

### 4.3 Experimental validation

To validate Theorems 4.1 and 4.2, we ran augmented GD and SGD with additive Gaussian noise on $N = 100$ simulated datapoints. Inputs were i.i.d. Gaussian vectors in dimension $n = 400$, and outputs in dim $p = 1$ were generated by a random linear map with i.i.d Gaussian coefficients drawn from $\mathcal{N}(1, 1)$. The learning rate followed a fixed polynomially decaying schedule $\eta_t = \frac{0.005}{100} \cdot (\text{batch size}) \cdot (1 + \frac{t}{20})^{-0.66}$, and the batch size used for SGD was 20. Figure 4.1 shows MSE and $\|W_{t,\perp}\|_F$ along a single optimization trajectory with different schedules for the variance $\sigma_t^2$ used in Gaussian noise augmentation. Complete code to generate this figure is provided in `supplement.zip` in the supplement. It ran in 30 minutes on a standard laptop CPU.

For both GD and SGD, Figure 4.1 shows that the optimization trajectory reaches $W_{\min}$ only when both learning rate and noise variance decay polynomially to zero. Indeed, Figure 4.1 shows that if $\sigma_t^2$ is zero (blue) or exponentially decaying (green), then while the MSE tends to zero, the orthogonal component $W_{t,\perp}$ does not tend to zero. Thus these choices of augmentation schedule cause $W_t$ to converge to an optimum which does not have minimal norm.

On the other hand, if $\sigma_t^2$ remains constant (orange), then while $W_{t,\perp}$ tends to zero, the MSE is not minimized. Only by decaying both noise strength and learning rate to 0 at sufficiently slow polynomial rates (red) prescribed by Theorem 4.1 do we find both MSE and $W_{t,\perp}$ tending to 0, meaning that augmented (S)GD finds the minimum norm optimum $W_{\min}$ under this choice of parameter scheduling.

## 5 Special Case: Augmentation with Random Projections

We further illustrate our results by specializing them to a class of augmentations which replace each input $x$ in a batch by its orthogonal projection $\Pi_t X$ onto a random subspace. In practice (e.g. when using CutOut [12] or SpecAugment [23]), the subspace is chosen based on a prior about correlations between components of $X$, but we consider the simplified case of a uniformly random subspace of $\mathbb{R}^n$ of given dimension.

At each time step $t$ we fix a dimension $k_t$ and a fixed $k_t$-dimensional subspace $\widetilde{S}_t$ of $\mathbb{R}^n$. Define the random subspace $S_t$ by

$$S_t := Q_t(\widetilde{S}_t) = \{Q_t x \mid x \in \widetilde{S}_t\},$$

where $Q_t \in O(n)$ is a Haar random orthogonal matrix. Thus, $S_t$ is uniformly distributed among all $k_t$-dimensional subspaces in $\mathbb{R}^n$. At step $t$, we take the augmentation given by

$$X_t = \Pi_t X \qquad Y_t = Y, \qquad \Pi_t := Q_t \widetilde{\Pi}_t Q_t^{\mathsf{T}},$$

where $\widetilde{\Pi}_t$ is the orthogonal projection onto $\widetilde{S}_t$ and hence $\Pi_t$ is the orthogonal projection onto $S_t$.

Denoting by $\gamma_t = k_t/n$ the relative dimension of $S_t$, a direct computation (see Lemma E.1) reveals that the proxy loss $\overline{\mathcal{L}}_t(W)$ equals $\mathcal{L}(\gamma_t W; \mathcal{D})$ plus

$$\frac{1}{N}\frac{\gamma_t(1-\gamma_t)}{n}\|X\|_F^2 \cdot \|W\|_F^2 + \frac{\gamma_t(1-\gamma_t)(1/n - 2/n^2)}{N(1+1/n-2/n^2)}(\|WX\|_F^2 + \frac{1}{n}\|X\|_F^2 \cdot \|W\|_F^2). \quad (5.1)$$

Neglecting terms of order $O(n^{-1})$, this proxy loss applies a Stein-type shrinkage on input data by $\gamma_t$ and adds a data-dependent $\ell_2$ penalty. For $\gamma_t < 1$, the minimizer of the proxy loss (5.1) is

$$W_{\gamma_t}^* = YX^\mathsf{T}\left(\frac{\gamma_t + 1/n - 2/n^2}{1+1/n-2/n^2}XX^\mathsf{T} + \frac{1-\gamma_t}{1+1/n-2/n^2}\frac{\|X\|_F^2}{n}\,\mathrm{Id}\right)^{-1}.$$

Again, although $W_{\gamma_t}^*$ does not minimize the original objective for any $\gamma_t < 1$, the sequence of these proxy optima converges to the minimal norm optimum in the weak regularization limit. Namely, we have $\lim_{\gamma_t \to 1^-} W_{\gamma_t}^* = W_{\min}$. Specializing our general result Theorem 3.1 to this setting, we obtain explicit conditions under which joint schedules of the normalized rank of the projection and the learning rate guarantee convergence to the minimum norm optimizer $W_{\min}$.

**Theorem 5.1.** *Suppose that $\eta_t \to 0, \gamma_t \to 1$ with $\gamma_t$ non-decreasing and*

$$\sum_{t=0}^\infty \eta_t(1-\gamma_t) = \infty \ \ and \ \ \sum_{t=0}^\infty \eta_t^2(1-\gamma_t) < \infty. \quad (5.2)$$

*Then, $W_t \xrightarrow{p} W_{min}$. Further, if $\eta_t = \Theta(t^{-x})$ and $\gamma_t = 1 - \Theta(t^{-y})$ with $x, y > 0$, $x + y < 1$, and $2x + y > 1$, then for small $\varepsilon > 0$, we have that $t^{\min\{y, \frac{1}{2}x\}-\varepsilon}\|W_t - W_{\min}\|_F \xrightarrow{p} 0$.*

Comparing the conditions (5.2) of Theorem 5.1 to the conditions (4.4) of Theorem 4.1, we see that $1 - \gamma_t$ is a measure of the strength of the random projection preconditioning. As in that setting, the fastest rates of convergence guaranteed by Theorem 5.1 are obtained by setting $\eta_t = t^{-2/3+\varepsilon}$ and $\gamma_t = 1 - t^{-1/3}$, yielding a $O(t^{-1/3+\varepsilon})$ rate of convergence.

## 6 Discussion and Limitations

We have presented a theoretical framework to rigorously analyze the effect of data augmentation. As can be seen in our main results, our framework applies to completely general augmentations and relies only on analyzing the first few moments of the augmented dataset. This allows us to handle augmentations as diverse as additive noise and random projections as well as their composition in a uniform manner. We have analyzed some representative examples in detail in this work, but many other commonly used augmentations may be handled similarly: label-preserving transformations (e.g. color jitter, geometric transformations) and Mixup [35], among many others. Another line of investigation left to future work is to compare different methods of combining augmentations such as mixing, alternating, or composing, which often improve performance in the empirical literature [16].

Though our results provide a rigorous baseline to compare to more complex settings, the restriction of the present work to linear models is a significant constraint. In future work, we hope to extend our general analysis to models closer to those used in practice. Most importantly, we intend to consider more complex models such as kernels (including the neural tangent kernel) and neural networks by making similar connections to stochastic optimization. In an orthogonal direction, our analysis currently focuses on the mean square loss for regression, and we aim to extend it to other losses such as cross-entropy. Finally, our study has thus far been restricted to the effect of data augmentation on optimization, and it would be of interest to derive consequences for generalization with more complex models. We hope our framework can provide the theoretical underpinnings for a more principled understanding of the effect and practice of data augmentation.

## Broader Impact

Our work provides a new theoretical approach to data augmentation for neural networks. By giving a better understanding of how this common practice affects optimization, we hope that it can lead to

more robust and interpretable uses of data augmentation in practice. Because our work is theoretical and generic, we do not envision negative impacts aside from those arising from improving learning algorithms in general.

## Acknowledgments and Disclosure of Funding

It is a pleasure to thank Daniel Park, Ethan Dyer, Edgar Dobriban, and Pokey Rule for a number of insightful conversations about data augmentation. B.H. was partially supported by NSF grants DMS-1855684 and DMS-2133806 and ONR MURI "Theoretical Foundations of Deep Learning." Y. S. was partially supported by NSF grants DMS-1701654/2039183 and DMS-2054838.

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
