# A Analytic lemmas

In this section, we present several basic lemmas concerning convergence for certain matrix-valued recursions that will be needed to establish our main results. For clarity, we first collect some matrix notations used in this section and throughout the paper.

## A.1 Matrix notations

Let $M \in \mathbb{R}^{m \times n}$ be a matrix. We denote its Frobenius norm by $\|M\|_F$ and its spectral norm by $\|M\|_2$. If $m = n$ so that $M$ is square, we denote by $\mathrm{diag}(M)$ the diagonal matrix with $\mathrm{diag}(M)_{ii} = M_{ii}$. For matrices $A, B, C$ of the appropriate shapes, define

$$A \circ (B \otimes C) := BAC \tag{A.1}$$

and

$$\mathrm{Var}(A) := \mathbb{E}[A^\mathsf{T} \otimes A] - \mathbb{E}[A^\mathsf{T}] \otimes \mathbb{E}[A]. \tag{A.2}$$

Notice in particular that

$$\mathrm{tr}[\mathrm{Id} \circ \mathrm{Var}(A)] = \mathbb{E}[\|A - \mathbb{E}[A]\|_F^2].$$

## A.2 One- and two-sided decay

**Definition A.1.** *Let $A_t \in \mathbb{R}^{n \times n}$ be a sequence of independent random non-negative definite matrices with*

$$\sup_t \|A_t\| \leq 2 \quad \textit{almost surely,}$$

*let $B_t \in \mathbb{R}^{p \times n}$ be a sequence of arbitrary matrices, and let $C_t \in \mathbb{R}^{n \times n}$ be a sequence of non-negative definite matrices. We say that the sequence of matrices $X_t \in \mathbb{R}^{p \times n}$ has one-sided decay of type $(\{A_t\}, \{B_t\})$ if it satisfies*

$$X_{t+1} = X_t(\mathrm{Id} - \mathbb{E}[A_t]) + B_t. \tag{A.3}$$

*We say that a sequence of non-negative definite matrices $Z_t \in \mathbb{R}^{n \times n}$ has two-sided decay of type $(\{A_t\}, \{C_t\})$ if it satisfies*

$$Z_{t+1} = \mathbb{E}[(\mathrm{Id} - A_t)Z_t(\mathrm{Id} - A_t)] + C_t. \tag{A.4}$$

Intuitively, if a sequence of matrices $X_t$ (resp. $Z_t$) satisfies one decay of type $(\{A_t\}, \{B_t\})$ (resp. two-sided decay of type $(\{A_t\}, \{C_t\})$), then in those directions $u \in \mathbb{R}^n$ for which $\|A_t u\|$ does not decay too quickly in $t$ we expect that $X_t$ (resp. $Z_t$) will converge to 0 provided $B_t$ (resp. $C_t$) are not too large. More formally, let us define

$$V_\| := \bigcap_{t=0}^\infty \ker\left[\prod_{s=t}^\infty (\mathrm{Id} - \mathbb{E}[A_s])\right] = \left\{ u \in \mathbb{R}^n \ \middle| \ \lim_{T \to \infty} \prod_{s=t}^T (\mathrm{Id} - \mathbb{E}[A_s])u = 0, \quad \forall t \geq 1 \right\},$$

and let $Q_\|$ be the orthogonal projection onto $V_\|$. It is on the space $V_\|$ that that we expect $X_t, Z_t$ to tend to zero if they satisfy one or two-side decay, and the precise results follows.

## A.3 Lemmas on Convergence for Matrices with One and Two-Sided Decay

We state here several results that underpin the proofs of our main results. We begin by giving in Lemmas A.2 and A.3 two slight variations of the same simple argument that matrices with one or two-sided decay converge to zero.

**Lemma A.2.** *If a sequence $\{X_t\}$ has one-sided decay of type $(\{A_t\}, \{B_t\})$ with*

$$\sum_{t=0}^\infty \|B_t\|_F < \infty, \tag{A.5}$$

*then $\lim_{t \to \infty} X_t Q_\| = 0$.*

*Proof.* For any $\varepsilon > 0$, choose $T_1$ so that $\sum_{t=T_1}^{\infty} \|B_t\|_F < \frac{\varepsilon}{2}$ and $T_2$ so that for $t > T_2$ we have

$$\left\| \Big( \prod_{s=T_1}^{t} (\mathrm{Id} - \mathbb{E}[A_s]) \Big) Q_{\|} \right\|_2 < \frac{\varepsilon}{2} \frac{1}{\|X_0\|_F + \sum_{s=0}^{T_1-1} \|B_s\|_F}.$$

By (A.3), we find that

$$X_{t+1} = X_0 \prod_{s=0}^{t} (\mathrm{Id} - \mathbb{E}[A_s]) + \sum_{s=0}^{t} B_s \prod_{r=s+1}^{t} (\mathrm{Id} - \mathbb{E}[A_r]),$$

which implies for $t > T_2$ that

$$\|X_{t+1} Q_{\|}\|_F \leq \|X_0\|_F \left\| \Big( \prod_{s=0}^{t} (\mathrm{Id} - \mathbb{E}[A_s]) \Big) Q_{\|} \right\|_2 + \sum_{s=0}^{t} \|B_s\|_F \left\| \Big( \prod_{r=s+1}^{t} (\mathrm{Id} - \mathbb{E}[A_r]) \Big) Q_{\|} \right\|_2.$$

$$(A.6)$$

Our assumption that $\|A_t\| \leq 2$ almost surely implies that for any $T \leq t$

$$\left\| \Big( \prod_{s=0}^{t} (\mathrm{Id} - \mathbb{E}[A_s]) \Big) Q_{\|} \right\|_2 \leq \left\| \Big( \prod_{s=0}^{T} (\mathrm{Id} - \mathbb{E}[A_s]) \Big) Q_{\|} \right\|_2$$

since each term in the product is non-negative-definite. Thus, we find

$$\|X_{t+1} Q_{\|}\|_F \leq \left[ \|X_0\|_F + \sum_{s=0}^{T_1-1} \|B_s\|_F \right] \left\| \Big( \prod_{s=T_1}^{t} (\mathrm{Id} - \mathbb{E}[A_s]) \Big) Q_{\|} \right\|_2 + \sum_{s=T_1}^{t} \|B_s\|_F < \varepsilon.$$

Taking $t \to \infty$ and then $\varepsilon \to 0$ implies that $\lim_{t \to \infty} X_t Q_{\|} = 0$, as desired. $\square$

**Lemma A.3.** *If a sequence $\{Z_t\}$ has two-sided decay of type $(\{A_t\}, \{C_t\})$ with*

$$\lim_{T \to \infty} \mathbb{E} \left[ \left\| \Big( \prod_{s=t}^{T} (\mathrm{Id} - A_s) \Big) Q_{\|} \right\|_2^2 \right] = 0 \quad \text{for all } t \geq 0 \tag{A.7}$$

*and*

$$\sum_{t=0}^{\infty} \mathrm{tr}(C_t) < \infty, \tag{A.8}$$

*then $\lim_{t \to \infty} Q_{\|}^{\mathsf{T}} Z_t Q_{\|} = 0$.*

*Proof.* The proof is essentially identical to that of Lemma A.2. That is, for $\varepsilon > 0$, choose $T_1$ so that $\sum_{t=T_1}^{\infty} \mathrm{tr}(C_t) < \frac{\varepsilon}{2}$ and choose $T_2$ by (A.7) so that for $t > T_2$ we have

$$\mathbb{E} \left[ \left\| \Big( \prod_{s=T_1}^{t} (\mathrm{Id} - A_s) \Big) Q_{\|} \right\|_2^2 \right] < \frac{\varepsilon}{2} \frac{1}{\mathrm{tr}(Z_0) + \sum_{s=0}^{T_1-1} \mathrm{tr}(C_s)}.$$

Conjugating (A.4) by $Q_{\|}$, we have that

$$Q_{\|}^{\mathsf{T}} Z_{t+1} Q_{\|} = \mathbb{E} \left[ Q_{\|}^{\mathsf{T}} \Big( \prod_{s=0}^{t} (\mathrm{Id} - A_s) \Big)^{\mathsf{T}} Z_0 \Big( \prod_{s=0}^{t} (\mathrm{Id} - A_s) \Big) Q_{\|} \right]$$

$$+ \sum_{s=0}^{t} \mathbb{E} \left[ Q_{\|}^{\mathsf{T}} \Big( \prod_{r=s+1}^{t} (\mathrm{Id} - A_r) \Big)^{\mathsf{T}} C_s \Big( \prod_{r=s+1}^{t} (\mathrm{Id} - A_r) \Big) Q_{\|} \right].$$

Our assumption that $\|A_t\| \leq 2$ almost surely implies that for any $T \leq t$

$$\left\| \Big( \prod_{s=0}^{t} (\mathrm{Id} - A_s) \Big) Q \right\|_2 \leq \left\| \Big( \prod_{s=0}^{T} (\mathrm{Id} - A_s) \Big) Q \right\|_2.$$

For $t > T_2$, this implies by taking trace of both sides that

$$\operatorname{tr}(Q_\parallel^\mathsf{T} Z_{t+1} Q_\parallel) \leq \operatorname{tr}(Z_0)\mathbb{E}\left[\left\|\left(\prod_{s=0}^{t}(\mathrm{Id} - A_s)\right)Q_\parallel\right\|_2^2\right] + \sum_{s=0}^{t}\operatorname{tr}(C_s)\mathbb{E}\left[\left\|\left(\prod_{r=s+1}^{t}(\mathrm{Id} - A_r)\right)Q_\parallel\right\|_2^2\right]$$

(A.9)

$$\leq \left[\operatorname{tr}(Z_0) + \sum_{s=0}^{T_1-1}\operatorname{tr}(C_s)\right]\mathbb{E}\left[\left\|\left(\prod_{s=T_1}^{t}(\mathrm{Id} - A_s)\right)Q_\parallel\right\|_2^2\right] + \sum_{s=T_1}^{t}\operatorname{tr}(C_s)$$

$$< \varepsilon,$$

which implies that $\lim_{t\to\infty} Q_\parallel^\mathsf{T} Z_t Q_\parallel = 0$.  $\qquad\square$

The preceding Lemmas will be used to provide sufficient conditions for augmented gradient descent to converge as in Theorem B.1 below. Since we are also interested in obtaining rates of convergence, we record here two quantitative refinements of the Lemmas above that will be used in the proof of Theorem B.4.

**Lemma A.4.** *Suppose $\{X_t\}$ has one-sided decay of type $(\{A_t\}, \{B_t\})$. Assume also that for some $X \geq 0$ and $C > 0$, we have*

$$\log\left\|\left(\prod_{r=s}^{t}(\mathrm{Id} - \mathbb{E}[A_r])\right)Q_\parallel\right\|_2 < X - C\int_s^{t+1} r^{-\alpha}\,dr$$

*and $\|B_t\|_F = O(t^{-\beta})$ for some $0 < \alpha < 1 < \beta$. Then, $\|X_t Q_\parallel\|_F = O(t^{\alpha-\beta})$.*

*Proof.* Denote $\gamma_{s,t} := \int_s^t r^{-\alpha}\,dr$. By (A.6), we have for some constants $C_1, C_2 > 0$ that

$$\|X_{t+1} Q_\parallel\|_F < C_1 e^{-C\gamma_{1,t+1}} + C_2 e^X \sum_{s=1}^{t}(1+s)^{-\beta} e^{-C\gamma_{s+1,t+1}}.$$

(A.10)

The first term on the right hand side is exponentially decaying in $t$ since $\gamma_{1,t+1}$ grows polynomially in $t$. To bound the second term, observe that the function

$$f(s) := C\gamma_{s+1,t+1} - \beta\log(s+1)$$

satisfies

$$f'(s) \geq 0 \quad \Leftrightarrow \quad C(s+1)^{-\alpha} - \frac{\beta}{1+s} \geq 0 \quad \Leftrightarrow \quad s \geq \left(\frac{\beta}{C}\right)^{1/(1-\alpha)} =: K.$$

Hence, the summands are monotonically increasing for $s$ greater than a fixed constant $K$ depending only on $\alpha, \beta, C$. Note that

$$\sum_{s=1}^{K}(1+s)^{-\beta} e^{-C\gamma_{s+1,t+1}} \leq K e^{-C\gamma_{K+1,t+1}} \leq K e^{-C't^{1-\alpha}}$$

for some $C'$ depending only on $\alpha$ and $K$, and hence sum is exponentially decaying in $t$. Further, using an integral comparison, we find

$$\sum_{s=K+1}^{t}(1+s)^{-\beta} e^{-C\gamma_{s+1,t+1}} \leq \int_K^t (1+s)^{-\beta} e^{-\frac{C}{1-\alpha}\left((t+1)^{1-\alpha}-(s+1)^{1-\alpha}\right)}\,ds.$$

(A.11)

Changing variables using $u = (1+s)^{1-\alpha}/(1-\alpha)$, the last integral has the form

$$e^{-Cg_t}(1-\alpha)^{-\xi}\int_{g_K}^{g_t} u^{-\xi} e^{Cu}\,du, \qquad g_x := \frac{(1+x)^{1-\alpha}}{1-\alpha}, \ \xi := \frac{\beta-\alpha}{1-\alpha}.$$

(A.12)

Integrating by parts, we have

$$\int_{g_K}^{g_t} u^{-\xi} e^u\,du = C^{-1}\xi\int_{g_K}^{g_t} u^{-\xi-1} e^{Cu}\,du + (u^{-\xi} e^{Cu})|_{g_K}^{g_t}$$

Further, since on the range $g_K \leq u \leq g_t$ the integrand is increasing, we have

$$e^{-Cg_t} \xi \int_{g_K}^{g_t} u^{-\xi-1} e^{Cu} du \leq \xi g_t^{-\xi}.$$

Hence, $e^{-Cg_t}$ times the integral in (A.12) is bounded above by

$$O(g_t^{-\xi}) + e^{-Cg_t} (u^{-\xi} e^{Cu})|_{g_K}^{g_t} = O(g_t^{-\xi}).$$

Using (A.11) and substituting the previous line into (A.12) yields the estimate

$$\sum_{s=K+1}^{t} (1+s)^{-\beta} e^{-C\gamma_{s+1,t+1}} \leq (1+t)^{-\beta+\alpha},$$

which completes the proof. $\qquad \square$

**Lemma A.5.** *Suppose $\{Z_t\}$ has two-sided decay of type $(\{A_t\}, \{C_t\})$. Assume also that for some $X \geq 0$ and $C > 0$, we have*

$$\log \mathbb{E} \left[ \left\| \left( \prod_{r=s}^{t} (\mathrm{Id} - A_r) \right) Q_\| \right\|_2^2 \right] < X - C \int_s^{t+1} r^{-\alpha} dr$$

*as well as $\mathrm{tr}(C_t) = O(t^{-\beta})$ for some $0 < \alpha < 1 < \beta$. Then $\mathrm{tr}(Q_\|^T Z_t Q_\|) = O(t^{\alpha-\beta})$.*

*Proof.* This argument is identical to the proof of Lemma A.4. Indeed, using (A.9) we have that

$$\mathrm{tr}\left( Q_\|^T Z_t Q_\| \right) \leq C_1 e^{-C\gamma_{1,t+1}} + C_2 e^X \sum_{s=1}^{t} (1+s)^{-\beta} e^{-C\gamma_{s+1,t+1}}.$$

The right hand side of this inequality coincides with the expression on the right hand side of (A.10), which we already bounded by $O(t^{\beta-\alpha})$ in the proof of Lemma A.4. $\qquad \square$

In what follows, we will use a concentration result for products of matrices from [18]. Let $Y_1, \ldots, Y_n \in \mathbb{R}^{N \times N}$ be independent random matrices. Suppose that

$$\|\mathbb{E}[Y_i]\|_2 \leq a_i \qquad \text{and} \qquad \mathbb{E}\left[ \|Y_i - \mathbb{E}[Y_i]\|_2^2 \right] \leq b_i^2 a_i^2$$

for some $a_1, \ldots, a_n$ and $b_1, \ldots, b_n$. We will use the following result, which is a specialization of Theorem 5.1 in [18] for $p = q = 2$.

**Theorem A.6** (Theorem 5.1 in [18])**.** *For $Z_0 \in \mathbb{R}^{N \times n}$, the product $Z_n = Y_n Y_{n-1} \cdots Y_1 Z_0$ satisfies*

$$\mathbb{E}\left[ \|Z_n\|_2^2 \right] \leq e^{\sum_{i=1}^n b_i^2} \prod_{i=1}^{n} a_i^2 \cdot \|Z_0\|_2^2$$

$$\mathbb{E}\left[ \|Z_n - \mathbb{E}[Z_n]\|_2^2 \right] \leq \left( e^{\sum_{i=1}^n b_i^2} - 1 \right) a_i^2 \cdot \|Z_0\|_2^2.$$

Finally, we collect two simple analytic lemmas for later use.

**Lemma A.7.** *For any matrix $M \in \mathbb{R}^{m \times n}$, we have that*

$$\mathbb{E}[\|M\|_2^2] \geq \|\mathbb{E}[M]\|_2^2.$$

*Proof.* We find by Cauchy-Schwartz and the convexity of the spectral norm that

$$\mathbb{E}[\|M\|_2^2] \geq \mathbb{E}[\|M\|_2]^2 \geq \|\mathbb{E}[M]\|_2^2. \qquad \square$$

**Lemma A.8.** *For bounded $a_t \geq 0$, if we have $\sum_{t=0}^{\infty} a_t = \infty$, then for any $C > 0$ we have*

$$\sum_{t=0}^{\infty} a_t e^{-C \sum_{s=0}^{t} a_s} < \infty.$$

*Proof.* Define $b_t := \sum_{s=0}^{t} a_s$ so that

$$S := \sum_{t=0}^{\infty} a_t e^{-C \sum_{s=0}^{t} a_s} = \sum_{t=0}^{\infty} (b_t - b_{t-1}) e^{-Cb_t} \leq \int_0^{\infty} e^{-Cx} dx < \infty,$$

where we use $\int_0^{\infty} e^{-Cx} dx$ to upper bound its right Riemann sum. $\qquad \square$

# B   Analysis of data augmentation as stochastic optimization

In this section, we prove generalizations of our main theoretical results Theorems 3.1 and 3.2 giving Monro-Robbins type conditions for convergence and rates for augmented gradient descent in the linear setting.

## B.1   Monro-Robbins type results

To state our general Monro-Robbins type convergence results, let us briefly recall the notation. We consider overparameterized linear regression with loss

$$\mathcal{L}(W; \mathcal{D}) = \frac{1}{N} \|WX - Y\|_F^2,$$

where the dataset $\mathcal{D}$ of size $N$ consists of data matrices $X, Y$ that each have $N$ columns $x_i \in \mathbb{R}^n, y_i \in \mathbb{R}^p$ with $n > N$. We optimize $\mathcal{L}(W; \mathcal{D})$ by augmented gradient descent, which means that at each time $t$ we replace $\mathcal{D} = (X, Y)$ by a random dataset $\mathcal{D}_t = (X_t, Y_t)$. We then take a step

$$W_{t+1} = W_t - \eta_t \nabla_W \mathcal{L}(W_t; \mathcal{D}_t)$$

of gradient descent on the resulting randomly augmented loss $\mathcal{L}(W; \mathcal{D}_t)$ with learning rate $\eta_t$. Recall that we set

$$V_\| := \text{ column span of } \mathbb{E}[X_t X_t^\mathsf{T}]$$

and denoted by $Q_\|$ the orthogonal projection onto $V_\|$. As noted in §3, on $V_\|$ the proxy loss

$$\overline{\mathcal{L}}_t = \mathbb{E}\left[\mathcal{L}(W; \mathcal{D}_t)\right]$$

is strictly convex and has a unique minimum, which is

$$W_t^* = \mathbb{E}\left[Y_t X_t^T\right] (Q_\| \mathbb{E}\left[X_t X_t^\mathsf{T}\right] Q_\|)^{-1}.$$

The change from one step of augmented GD to the next in these proxy optima is captured by

$$\Xi_t^* := W_{t+1}^* - W_t^*.$$

With this notation, we are ready to state Theorems B.1, which gives two different sets of time-varying Monro-Robbins type conditions under which the optimization trajectory $W_t$ converges for large $t$. In Theorem B.4, we refine the analysis to additionally give rates of convergence. Note that Theorem B.1 is a generalization of Theorem 3.1 and that Theorem B.4 is a generalization of Theorem 3.2.

**Theorem B.1.** *Suppose that $V_\|$ is independent of $t$, that the learning rate satisfies $\eta_t \to 0$, that the proxy optima satisfy*

$$\sum_{t=0}^{\infty} \|\Xi_t^*\|_F < \infty, \tag{B.1}$$

*ensuring the existence of a limit $W_\infty^* := \lim_{t \to \infty} W_t^*$ and that*

$$\sum_{t=0}^{\infty} \eta_t \lambda_{min, V_\|}\left(\mathbb{E}[X_t X_t^\mathsf{T}]\right) = \infty. \tag{B.2}$$

*Then if either*

$$\sum_{t=0}^{\infty} \eta_t^2 \mathbb{E}\left[\|X_t X_t^\mathsf{T} - \mathbb{E}[X_t X_t^\mathsf{T}]\|_F^2 + \|Y_t X_t^\mathsf{T} - \mathbb{E}[Y_t X_t^\mathsf{T}]\|_F^2\right] < \infty \tag{B.3}$$

*or*

$$\sum_{t=0}^{\infty} \eta_t^2 \mathbb{E}\left[\|X_t X_t^\mathsf{T} - \mathbb{E}[X_t X_t^\mathsf{T}]\|_F^2 + \left\|\mathbb{E}[W_t](X_t X_t^\mathsf{T} - \mathbb{E}[X_t X_t^\mathsf{T}]) - (Y_t X_t^\mathsf{T} - \mathbb{E}[Y_t X_t^\mathsf{T}])\right\|_F^2\right] < \infty \tag{B.4}$$

*hold, then for any initialization $W_0$, we have $W_t Q_\| \xrightarrow{p} W_\infty^*$.*

**Remark B.2.** *In the general case, the column span $V_{||}$ of $\mathbb{E}[X_t X_t^{\mathsf{T}}]$ may vary with $t$. This means that some directions in $\mathbb{R}^n$ may only have non-zero overlap with $colspan(\mathbb{E}[X_t X_t^{\mathsf{T}}])$ for some positive but finite collection of values of $t$. In this case, only finitely many steps of the optimization would move $W_t$ in this direction, meaning that we must define a smaller space for convergence. The correct definition of this subspace turns out to be the following*

$$V_{||} := \bigcap_{t=0}^{\infty} \ker\left[ \prod_{s=t}^{\infty} \left( \mathrm{Id} - \frac{2\eta_s}{N} \mathbb{E}[X_s X_s^{\mathsf{T}}] \right) \right] \tag{B.5}$$

$$= \bigcap_{t=0}^{\infty} \left\{ u \in \mathbb{R}^n \;\middle|\; \lim_{T\to\infty} \prod_{s=t}^{T} \left( \mathrm{Id} - \frac{2\eta_s}{N} \mathbb{E}[X_s X_s^{\mathsf{T}}] \right) u = 0 \right\}.$$

*With this re-definition of $V_{||}$ and with $Q_{||}$ still denoting the orthogonal projection to $V_{||}$, Theorem B.1 holds verbatim and with the same proof. Note that if $\eta_t \to 0$, $V_{||} = colspan(\mathbb{E}[X_t X_t^{\mathsf{T}}])$ is fixed in $t$, and (B.2) holds, this definition of $V_{||}$ reduces to that defined in (3.5).*

**Remark B.3.** *The condition (B.4) can be written in a more conceptual way as*

$$\sum_{t=0}^{\infty} \left[ \|X_t X_t^{\mathsf{T}} - \mathbb{E}[X_t X_t^{\mathsf{T}}]\|_F^2 + \eta_t^2 tr\left[ \mathrm{Id} \circ Var\left( (\mathbb{E}[W_t]X_t - Y_t)X_t^{\mathsf{T}} \right) \right] \right] < \infty,$$

*where we recognize that $(\mathbb{E}[W_t]X_t - Y_t)X_t^{\mathsf{T}}$ is precisely the stochastic gradient estimate at time $t$ for the proxy loss $\overline{\mathcal{L}}_t$, evaluated at $\mathbb{E}[W_t]$, which is the location at time $t$ for vanilla GD on $\overline{\mathcal{L}}_t$ since taking expectations in the GD update equation (3.3) coincides with GD for $\overline{\mathcal{L}}_t$. Moreover, condition (B.4) actually implies condition (B.3) (see (B.12) below). The reason we state Theorem B.1 with both conditions, however, is that (B.4) makes explicit reference to the average $\mathbb{E}[W_t]$ of the augmented trajectory. Thus, when applying Theorem B.1 with this weaker condition, one must separately estimate the behavior of this quantity.*

Theorem B.1 gave conditions on joint learning rate and data augmentation schedules under which augmented optimization is guaranteed to converge. Our next result proves rates for this convergence.

**Theorem B.4.** *Suppose that $\eta_t \to 0$ and that for some $0 < \alpha < 1 < \beta_1, \beta_2$ and $C_1, C_2 > 0$, we have*

$$\log \mathbb{E}\left[ \left\| \left( \prod_{r=s}^{t} \left( \mathrm{Id} - \frac{2\eta_r}{N} X_r X_r^{\mathsf{T}} \right) \right) Q_{||} \right\|_2^2 \right] < C_1 - C_2 \int_s^{t+1} r^{-\alpha} dr \tag{B.6}$$

*as well as*

$$\|\Xi_t^*\|_F = O(t^{-\beta_1}) \tag{B.7}$$

*and*

$$\eta_t^2 tr\left[ \mathrm{Id} \circ Var(\mathbb{E}[W_t]X_t X_t^{\mathsf{T}} - Y_t X_t^{\mathsf{T}}) \right] = O(t^{-\beta_2}). \tag{B.8}$$

*Then, for any initialization $W_0$, we have for any $\varepsilon > 0$ that*

$$t^{\min\{\beta_1 - 1, \frac{\beta_2 - \alpha}{2}\} - \varepsilon} \|W_t Q_{||} - W_\infty^*\|_F \xrightarrow{p} 0.$$

**Remark B.5.** *To reduce Theorem 3.2 to Theorem B.4, we notice that (3.10) and (3.11) mean that Theorem A.6 applies to $Y_t = \mathrm{Id} - 2\eta_t \frac{X_t X_t^{\mathsf{T}}}{N}$ with $a_t = 1 - \Omega(t^{-\alpha})$ and and $b_t^2 = O(t^{-\gamma})$, thus implying (B.6).*

The first step in proving both Theorem B.1 and Theorem B.4 is to obtain recursions for the mean and variance of the difference $W_t - W_t^*$ between the time $t$ proxy optimum and the augmented optimization trajectory at time $t$. We will then complete the proof of Theorem B.1 in §B.3 and the proof of Theorem B.4 in §B.4.

## B.2 Recursion relations for parameter moments

The following proposition shows that difference between the mean augmented dynamics $\mathbb{E}[W_t]$ and the time$-t$ optimum $W_t^*$ satisfies, in the sense of Definition A.1, one-sided decay of type $(\{A_t\}, \{B_t\})$ with

$$A_t = \frac{2\eta_t}{N} X_t X_t^{\mathsf{T}}, \qquad B_t = -\Xi_t^*.$$

It also shows that the variance of this difference, which is non-negative definite, satisfies two-sided decay of type $(\{A_t\}, \{C_t\})$ with $A_t$ as before and

$$C_t = \frac{4\eta_t^2}{N^2}\left[\text{Id} \circ \text{Var}\left(\mathbb{E}[W_t]X_tX_t^\mathsf{T} - Y_tX_t^\mathsf{T}\right)\right].$$

In terms of the notations of Appendix A.1, we have the following recursions.

**Proposition B.6.** *The quantity* $\mathbb{E}[W_t] - W_t^*$ *satisfies*

$$\mathbb{E}[W_{t+1}] - W_{t+1}^* = (\mathbb{E}[W_t] - W_t^*)\left(\text{Id} - \frac{2\eta_t}{N}\mathbb{E}[X_tX_t^\mathsf{T}]\right) - \Xi_t^* \tag{B.9}$$

*and* $Z_t := \mathbb{E}[(W_t - \mathbb{E}[W_t])^\mathsf{T}(W_t - \mathbb{E}[W_t])]$ *satisfies*

$$Z_{t+1} = \mathbb{E}\left[(\text{Id} - \frac{2\eta_t}{N}X_tX_t^\mathsf{T})Z_t(\text{Id} - \frac{2\eta_t}{N}X_tX_t^\mathsf{T})\right] + \frac{4\eta_t^2}{N^2}\left[\text{Id} \circ \text{Var}\left(\mathbb{E}[W_t]X_tX_t^\mathsf{T} - Y_tX_t^\mathsf{T}\right)\right].$$
$$\tag{B.10}$$

*Proof.* Notice that $\mathbb{E}[X_tX_t^\mathsf{T}]u = 0$ if and only if $X_t^\mathsf{T}u = 0$ almost surely, which implies that

$$W_t^*\mathbb{E}[X_tX_t^\mathsf{T}] = \mathbb{E}[Y_tX_t^\mathsf{T}]\mathbb{E}[X_tX_t^\mathsf{T}]^+\mathbb{E}[X_tX_t^\mathsf{T}] = \mathbb{E}[Y_tX_t^\mathsf{T}].$$

Thus, the learning dynamics (3.3) yield

$$\mathbb{E}[W_{t+1}] = \mathbb{E}[W_t] - \frac{2\eta_t}{N}\left(\mathbb{E}[W_t]\mathbb{E}[X_tX_t^\mathsf{T}] - \mathbb{E}[Y_tX_t^\mathsf{T}]\right)$$

$$= \mathbb{E}[W_t] - \frac{2\eta_t}{N}(\mathbb{E}[W_t] - W_t^*)\mathbb{E}[X_tX_t^\mathsf{T}].$$

Subtracting $W_{t+1}^*$ from both sides yields (B.9). We now analyze the fluctuations. Writing $\text{Sym}(A) := A + A^\mathsf{T}$, we have

$$\mathbb{E}[W_{t+1}]^\mathsf{T}\mathbb{E}[W_{t+1}] = \mathbb{E}[W_t]^\mathsf{T}\mathbb{E}[W_t] + \frac{2\eta_t}{N}\text{Sym}\left(\mathbb{E}[W_t]^\mathsf{T}\mathbb{E}[Y_tX_t^\mathsf{T}] - \mathbb{E}[W_t]^\mathsf{T}\mathbb{E}[W_t]\mathbb{E}[X_tX_t^\mathsf{T}]\right)$$

$$+ \frac{4\eta_t^2}{N^2}\left(\mathbb{E}[X_tX_t^\mathsf{T}]\mathbb{E}[W_t]^\mathsf{T}\mathbb{E}[W_t]\mathbb{E}[X_tX_t^\mathsf{T}] + \mathbb{E}[X_tY_t^\mathsf{T}]\mathbb{E}[Y_tX_t^\mathsf{T}] - \text{Sym}(\mathbb{E}[X_tX_t^\mathsf{T}]\mathbb{E}[W_t]^\mathsf{T}\mathbb{E}[Y_tX_t^\mathsf{T}])\right).$$

Similarly, we have that

$$\mathbb{E}[W_{t+1}^\mathsf{T}W_{t+1}] = \mathbb{E}[W_t^\mathsf{T}W_t] + \frac{2\eta_t}{N}\text{Sym}(\mathbb{E}[W_t^\mathsf{T}Y_tX_t^\mathsf{T} - W_t^\mathsf{T}W_tX_tX_t^\mathsf{T}])$$

$$+ \frac{4\eta_t^2}{N^2}\mathbb{E}[X_tX_t^\mathsf{T}W_t^\mathsf{T}W_tX_tX_t^\mathsf{T} - \text{Sym}(X_tX_t^\mathsf{T}W_t^\mathsf{T}Y_tX_t^\mathsf{T}) + X_tY_t^\mathsf{T}Y_tX_t^\mathsf{T}].$$

Noting that $X_t$ and $Y_t$ are independent of $W_t$ and subtracting yields the desired. $\square$

### B.3  Proof of Theorem B.1

First, by Proposition B.6, we see that $\mathbb{E}[W_t] - W_t^*$ has one-sided decay with

$$A_t = 2\eta_t\frac{X_tX_t^\mathsf{T}}{N} \qquad \text{and} \qquad B_t = -\Xi_t^*.$$

Thus, by Lemma A.2 and (B.1), we find that

$$\lim_{t\to\infty}(\mathbb{E}[W_t]Q_\| - W_t^*) = 0, \tag{B.11}$$

which gives convergence in expectation.

For the second moment, by Proposition B.6, we see that $Z_t$ has two-sided decay with

$$A_t = 2\eta_t\frac{X_tX_t^\mathsf{T}}{N} \qquad \text{and} \qquad C_t = \frac{4\eta_t^2}{N^2}\left[\text{Id} \circ \text{Var}\left(\mathbb{E}[W_t]X_tX_t^\mathsf{T} - Y_tX_t^\mathsf{T}\right)\right].$$

We now verify (A.7) and (A.8) in order to apply Lemma A.3.

For (A.7), for any $\varepsilon > 0$, notice that

$$\mathbb{E}[\|A_s - \mathbb{E}[A_s]\|_F^2] = \eta_s^2 \mathbb{E}[\|X_s X_s^\mathsf{T} - \mathbb{E}[X_s X_s^\mathsf{T}]\|_F^2]$$

so by either (B.3) or (B.4) we may choose $T_1 > t$ so that $\sum_{s=T_1}^\infty \mathbb{E}[\|A_s - \mathbb{E}[A_s]\|_F^2] < \frac{\varepsilon}{2}$. Now choose $T_2 > T_1$ so that for $T > T_2$, we have

$$\left\|\left(\prod_{r=T_1}^{T} \mathbb{E}[\mathrm{Id} - A_r]\right)Q_\|\right\|_2^2 < \frac{\varepsilon}{2} \frac{1}{\|\prod_{s=t}^{T_1-1}\mathbb{E}[\mathrm{Id} - A_s]\|_F^2 + \sum_{s=t}^{T_1-1}\mathbb{E}[\|A_s - \mathbb{E}[A_s]\|_F^2]}.$$

For $T > T_2$, we then have

$$\mathbb{E}\left[\left\|\left(\prod_{s=t}^{T}(\mathrm{Id} - A_s)\right)Q_\|\right\|_2^2\right]$$

$$\leq \left\|\left(\prod_{s=t}^{T}\mathbb{E}[\mathrm{Id} - A_s]\right)Q_\|\right\|^2 + \sum_{s=t}^{T}\mathbb{E}\left[\left\|\prod_{r=t}^{s}(\mathrm{Id} - A_r)\prod_{r=s+1}^{T}(\mathrm{Id} - \mathbb{E}[A_r])Q_\|\right\|_F^2 - \left\|\prod_{r=t}^{s-1}(\mathrm{Id} - A_r)\prod_{r=s}^{T}(\mathrm{Id} - \mathbb{E}[A_r])Q_\|\right\|_F^2\right]$$

$$= \left\|\left(\prod_{s=t}^{T}\mathbb{E}[\mathrm{Id} - A_s]\right)Q_\|\right\|_F^2 + \sum_{s=t}^{T}\mathbb{E}\left[\left\|\prod_{r=t}^{s-1}(\mathrm{Id} - A_r)(A_s - \mathbb{E}[A_s])\prod_{r=s+1}^{T}(\mathrm{Id} - \mathbb{E}[A_r])Q_\|\right\|_F^2\right]$$

$$\leq \left\|\prod_{s=t}^{T_1-1}\mathbb{E}[\mathrm{Id} - A_s]\right\|_F^2 \left\|\left(\prod_{r=T_1}^{T}\mathbb{E}[\mathrm{Id} - A_r]\right)Q_\|\right\|_2^2 + \sum_{s=t}^{T}\mathbb{E}[\|A_s - \mathbb{E}[A_s]\|_F^2]\left\|\left(\prod_{r=s+1}^{T}\mathbb{E}[\mathrm{Id} - A_r]\right)Q_\|\right\|_2^2$$

$$\leq \left(\left\|\prod_{s=t}^{T_1-1}\mathbb{E}[\mathrm{Id} - A_s]\right\|_F^2 + \sum_{s=t}^{T_1-1}\mathbb{E}[\|A_s - \mathbb{E}[A_s]\|_F^2]\right)\left\|\left(\prod_{r=T_1}^{T}\mathbb{E}[\mathrm{Id} - A_r]\right)Q_\|\right\|_2^2 + \sum_{s=T_1}^{T}\mathbb{E}[\|A_s - \mathbb{E}[A_s]\|_F^2]$$

$$< \varepsilon,$$

which implies (A.7). Condition (A.8) follows from either (B.4) or (B.3) and the bounds

$$\mathrm{tr}(C_t) \leq \frac{8\eta_t^2}{N^2}\left(\|\mathbb{E}[W_t](X_t X_t^\mathsf{T} - \mathbb{E}[X_t X_t^\mathsf{T}])\|_F^2 + \|Y_t X_t^\mathsf{T} - \mathbb{E}[Y_t X_t^\mathsf{T}]\|_F^2\right) \tag{B.12}$$

$$\leq \frac{8\eta_t^2}{N^2}\left(\|\mathbb{E}[W_t]\|^2\|X_t X_t^\mathsf{T} - \mathbb{E}[X_t X_t^\mathsf{T}]\|_F^2 + \|Y_t X_t^\mathsf{T} - \mathbb{E}[Y_t X_t^\mathsf{T}]\|_F^2\right),$$

where in the first inequality we use the fact that $\|M_1 - M_2\|_F^2 \leq 2(\|M_1\|_F^2 + \|M_2\|_F^2)$. Furthermore, iterating (B.9) yields $\|\mathbb{E}[W_t] - W_t^*\|_F \leq \|W_0 - W_0^*\|_F + \sum_{t=0}^\infty \|\Xi_t^*\|_F$, which combined with (B.12) and either (B.3) or (B.4) therefore implies (A.8). We conclude by Lemma A.3 that

$$\lim_{t\to\infty} Q_\|^\mathsf{T} Z_t Q_\| = \lim_{t\to\infty} \mathbb{E}[Q_\|^\mathsf{T}(W_t - \mathbb{E}[W_t])^\mathsf{T}(W_t - \mathbb{E}[W_t])Q_\|] = 0. \tag{B.13}$$

Together, (B.11) and (B.13) imply that $W_t Q_\| - W_t^* \xrightarrow{p} 0$. The conclusion then follows from the fact that $\lim_{t\to 0} W_t^* = W_\infty^*$. This complete the proof of Theorem B.1. $\square$

## B.4 Proof of Theorem B.4

By Proposition B.6, $\mathbb{E}[W_t] - W_t^*$ has one-sided decay with

$$A_t = \frac{2\eta_t}{N}X_t X_t^\mathsf{T}, \qquad B_t = -\Xi_t^*.$$

By Lemma A.7 and (B.6), $\mathbb{E}[A_t]$ satisfies

$$\log\left\|\prod_{r=s}^{t}\left(\mathrm{Id} - 2\eta_r \frac{1}{N}\mathbb{E}[X_r X_r^\mathsf{T}]\right)Q_\|\right\|_2 \leq \frac{1}{2}\log\mathbb{E}\left[\left\|\left(\prod_{r=s}^{t}\left(\mathrm{Id} - 2\eta_r \frac{X_r X_r^\mathsf{T}}{N}\right)\right)Q_\|\right\|_2^2\right]$$

$$< \frac{C_1}{2} - \frac{C_2}{2}\int_s^{t+1} r^{-\alpha}dr.$$

Applying Lemma A.4 using this bound and (B.7), we find that

$$\|\mathbb{E}[W_t]Q_\| - W_t^*\|_F = O(t^{\alpha-\beta_1}).$$

Moreover, because $\|\Xi_t^*\|_F = O(t^{-\beta_1})$, we also find that $\|W_t^* - W_\infty^*\|_F = O(t^{-\beta_1+1})$, and hence

$$\|\mathbb{E}[W_t]Q_\| - W_\infty^*\|_F = O(t^{-\beta_1+1}).$$

Further, by Proposition B.6, $\mathbb{E}[(W_t - \mathbb{E}[W_t])^\mathsf{T}(W_t - \mathbb{E}[W_t])]$ has two-sided decay with

$$A_t = \frac{2\eta_t}{N}X_tX_t^\mathsf{T}, \qquad C_t = \frac{4\eta_t^2}{N^2}\left[\mathrm{Id}\circ\mathrm{Var}\left(\mathbb{E}[W_t]X_tX_t^\mathsf{T} - Y_tX_t^\mathsf{T}\right)\right].$$

Applying Lemma A.5 with (B.6) and (B.8), we find that

$$\mathbb{E}\left[\|(W_t - \mathbb{E}[W_t])Q_\|\|_F^2\right] = O(t^{\alpha-\beta_2}).$$

By Chebyshev's inequality, for any $x > 0$ we have

$$\mathbb{P}\left(\|W_tQ_\| - W_\infty^*\|_F \geq O(t^{-\beta_1+1}) + x\cdot O(t^{\frac{\alpha-\beta_2}{2}})\right) \leq x^{-2}.$$

For any $\varepsilon > 0$, choosing $x = t^\delta$ for small $0 < \delta < \varepsilon$ we find as desired that

$$t^{\min\{\beta_1-1, \frac{\beta_2-\alpha}{2}\}-\varepsilon}\|W_tQ_\| - W_\infty^*\|_F \xrightarrow{P} 0,$$

thus completing the proof of Theorem B.4. $\qquad\square$

## C  Intrinsic time

Theorem 3.2 measures rates in terms of optimization steps $t$, but a different measurement of time called the *intrinsic time* of the optimization will be more suitable for measuring the behavior of optimization quantities. This was introduced for SGD in [28, 27], and we now generalize it to our broader setting. For gradient descent on a loss $\mathcal{L}$, the intrinsic time is a quantity which increments by $\eta\lambda_{\min}(H)$ for a optimization step with learning rate $\eta$ at a point where $\mathcal{L}$ has Hessian $H$. When specialized to our setting, it is given by

$$\tau(t) := \sum_{s=0}^{t-1}\frac{2\eta_s}{N}\lambda_{\min,V_\|}(\mathbb{E}[X_sX_s^\mathsf{T}]). \tag{C.1}$$

Notice that intrinsic time of augmented optimization for the sequence of proxy losses $\overline{\mathcal{L}}_s$ appears in Theorems 3.1 and 3.2, which require via conditions (3.8) and (3.11) that the intrinsic time tends to infinity as the number of optimization steps grows.

Intrinsic time will be a sensible variable in which to measure the behavior of quantities such as the fluctuations of the optimization path $f(t) := \mathbb{E}[\|(W_t - \mathbb{E}[W_t])Q_\|\|_F^2]$. In the proofs of Theorems 3.1 and 3.2, we show that the fluctuations satisfy an inequality of the form

$$f(t+1) \leq f(t)(1 - a(t))^2 + b(t) \tag{C.2}$$

for $a(t) := 2\eta_t\frac{1}{N}\lambda_{\min,V_\|}(\mathbb{E}[X_tX_t^\mathsf{T}])$ and $b(t) := \mathrm{Var}[\|\eta_t\nabla_W\mathcal{L}(W_t)\|_F]$ so that $\tau(t) = \sum_{s=0}^{t-1}a(s)$. Iterating the recursion (C.2) shows that

$$f(t) \leq f(0)\prod_{s=0}^{t-1}(1 - a(s))^2 + \sum_{s=0}^{t-1}b(s)\prod_{r=s+1}^{t-1}(1 - a(r))^2$$

$$\leq e^{-2\tau(t)}f(0) + \sum_{s=0}^{t-1}\frac{b(s)}{a(s)}e^{2\tau(s+1)-2\tau(t)}(\tau(s+1) - \tau(s)).$$

Changing variables to $\tau := \tau(t)$ and defining $A(\tau)$, $B(\tau)$, and $F(\tau)$ by $A(\tau(t)) = a(t)$, $B(\tau(t)) = b(t)$, and $F(\tau(t)) = f(t)$, we find by replacing a right Riemann sum by an integral that

$$F(\tau) \precsim e^{-2\tau}\left[F(0) + \int_0^\tau\frac{B(\sigma)}{A(\sigma)}e^{2\sigma}d\sigma\right]. \tag{C.3}$$

In order for the result of optimization to be independent of the starting point, by (C.3) we must have $\tau \to \infty$ to remove the dependence on $F(0)$; this provides one explanation for the appearance of $\tau$ in condition (3.8). Further, (C.3) implies that the fluctuations at an intrinsic time are bounded by an integral against the function $\frac{B(\sigma)}{A(\sigma)}$ which depends only on the ratio of $A(\sigma)$ and $B(\sigma)$.

In the case of minibatch SGD, our proof of Theorem F.1 shows the intrinsic time is $\tau(t) = \sum_{s=0}^{t-1} 2\eta_s \frac{1}{N} \lambda_{\min, V_\parallel}(XX^\mathsf{T})$ and the ratio $\frac{b(t)}{a(t)}$ in (C.3) is by (F.5) bounded uniformly by $\frac{b(t)}{a(t)} \le C \cdot \frac{\eta_t}{B_t}$ for a constant $C > 0$. Thus, keeping $\frac{b(t)}{a(t)}$ fixed as a function of $\tau$ suggests the "linear scaling" $\eta_t \propto B_t$ used empirically in [13] and proposed via an heuristic SDE limit in [27].

# D   Analysis of Noising Augmentations

In this section, we give a full analysis of the noising augmentations presented in Section 4. Let us briefly recall the notation. As before, we consider overparameterized linear regression with loss

$$\mathcal{L}(W; \mathcal{D}) = \frac{1}{N} \|WX - Y\|_F^2,$$

where the dataset $\mathcal{D}$ of size $N$ consists of data matrices $X, Y$ that each have $N$ columns $x_i \in \mathbb{R}^n$, $y_i \in \mathbb{R}^p$ with $n > N$. We optimize $\mathcal{L}(W; \mathcal{D})$ by augmented gradient descent or augmented stochastic gradient descent with additive Gaussian noise. This means that at each time $t$ we replace $\mathcal{D} = (X, Y)$ by a random batch $\mathcal{D}_t = (X_t, Y)$ of size $B_t$, where the columns $x_{i,t}$ of $X_t$ are

$$x_{i,t} = x_i + \sigma_t G_{i,t}, \qquad G_{i,t} \sim \mathcal{N}(0, 1) \text{ i.i.d.}$$

In the case of gradient descent, the batch consists of the entire dataset, and the resulting data matrices are

$$X_t = X + \sigma_t G_t \qquad \text{and} \qquad Y_t = Y.$$

In the case of stochastic gradient descent, the batch consists of $B_t$ datapoints chosen uniformly at random with replacement, and the resulting data matrices are

$$X_t = c_t(XA_t + \sigma_t G_t) \qquad \text{and} \qquad Y_t = c_t Y A_t,$$

where $c_t = \sqrt{N/B_t}$, $A_t \in \mathbb{R}^{N \times B_t}$ has i.i.d. columns with a single non-zero entry equal to 1, and $G_t \in \mathbb{R}^{n \times B_t}$ has i.i.d. Gaussian entries. In both cases, the proxy loss is

$$\overline{\mathcal{L}}_t(W) := \frac{1}{N} \|Y - WX\|_F^2 + \sigma_t^2 \|W\|_F^2,$$

which has ridge minimizer

$$W_t^* = YX^\mathsf{T}(XX^\mathsf{T} + \sigma_t^2 N \cdot \mathrm{Id}_{n \times n})^{-1},$$

and the space $V_\parallel :=$ column span of $\mathbb{E}[X_t X_t^\mathsf{T}]$ is all of $\mathbb{R}^n$. We now separately analyze the cases of GD and SGD in Theorems 4.1 and Theorem 4.2, respectively.

## D.1   Proof of Theorem 4.1 for GD

We begin by proving convergence without rates. For this, we seek to show that if $\sigma_t^2, \eta_t \to 0$ with $\sigma_t^2$ non-increasing and

$$\sum_{t=0}^{\infty} \eta_t \sigma_t^2 = \infty \qquad \text{and} \qquad \sum_{t=0}^{\infty} \eta_t^2 \sigma_t^2 < \infty, \tag{D.1}$$

then, $W_t \xrightarrow{p} W_{\min}$. We will do this by applying Theorem 3.1, so we check that our assumptions imply the hypotheses of these theorems. For Theorem 3.1, we directly compute

$$\mathbb{E}[Y_t X_t^\mathsf{T}] = YX^\mathsf{T} \qquad \text{and} \qquad \mathbb{E}[X_t X_t^\mathsf{T}] = XX^\mathsf{T} + \sigma_t^2 N \cdot \mathrm{Id}_{n \times n}$$

and

$$\mathbb{E}[X_t X_t^\mathsf{T} X_t] = XX^\mathsf{T} X + \sigma_t^2(N + n + 1)X$$

$$\mathbb{E}[X_t X_t^\mathsf{T} X_t X_t^\mathsf{T}] = XX^\mathsf{T} XX^\mathsf{T} + \sigma_t^2\Big((2N + n + 2)XX^\mathsf{T} + \mathrm{tr}(XX^\mathsf{T})\,\mathrm{Id}_{n \times n}\Big) + \sigma_t^4 N(N + n + 1)\,\mathrm{Id}_{n \times n}.$$

We also find that

$$\|\Xi_t^*\|_F = |\sigma_t^2 - \sigma_{t+1}^2|N \left\| Y X^\mathsf{T} \Big( X X^\mathsf{T} + \sigma_t^2 N \cdot \mathrm{Id}_{n\times n} \Big)^{-1} \Big( X X^\mathsf{T} + \sigma_{t+1}^2 N \cdot \mathrm{Id}_{n\times n} \Big)^{-1} \right\|_F$$

$$\leq |\sigma_t^2 - \sigma_{t+1}^2|N\|Y X^\mathsf{T}[(X X^\mathsf{T})^+]^2\|_F.$$

Thus, because $\sigma_t^2$ is decreasing, we see that the hypothesis (3.7) of Theorem 3.1 indeed holds. Further, we note that

$$\sum_{t=0}^\infty \eta_t^2 \mathbb{E}\left[\|X_t X_t^\mathsf{T} - \mathbb{E}[X_t X_t^\mathsf{T}]\|_F^2 + \|Y_t X_t^\mathsf{T} - \mathbb{E}[Y_t X_t^\mathsf{T}]\|_F^2\right]$$

$$= \sum_{t=0}^\infty \eta_t^2 \sigma_t^2 \Big( 2(n+1)\|X\|_F^2 + N\|Y\|_F^2 + \sigma_t^2 N n(n+1) \Big) = O\left( \sum_{t=0}^\infty \eta_t^2 \sigma_t^2 \right),$$

which by (D.1) implies (B.3). Theorem 3.1 and the fact that $\lim_{t\to\infty} W_t^* = W_{\min}$ therefore yield that $W_t \xrightarrow{p} W_{\min}$.

We now prove convergence with rates, we aim to show that if $\eta_t = \Theta(t^{-x})$ and $\sigma_t^2 = \Theta(t^{-y})$ with $x, y > 0$, $x + y < 1$, and $2x + y > 1$, then for any $\varepsilon > 0$, we have that

$$t^{\min\{y, \frac{1}{2}x\} - \varepsilon} \|W_t - W_{\min}\|_F \xrightarrow{p} 0.$$

We now check the hypotheses for and apply Theorem B.4. For (B.6), notice that $Y_r = \mathrm{Id} - 2\eta_r \frac{X_r X_r^\mathsf{T}}{N}$ satisfies the hypotheses of Theorem A.6 with $a_r = 1 - 2\eta_r \sigma_r^2$ and $b_r^2 = \frac{\eta_r^2 \sigma_r^2}{a_r^2}\Big( 2(n+1)\|X\|_F^2 + \sigma_r^2 N n(n+1) \Big)$. Thus, by Theorem A.6 and the fact that $\eta_t = \Theta(t^{-x})$ and $\sigma_t^2 = \Theta(t^{-y})$, we find for some $C_1, C_2 > 0$ that

$$\log \mathbb{E}\left[\left\| \prod_{r=s}^t (\mathrm{Id} - 2\eta_r \frac{X_r X_r^\mathsf{T}}{N}) \right\|_2^2 \right] \leq \sum_{r=s}^t b_r^2 + 2\sum_{r=s}^t \log(1 - 2\eta_r \sigma_r^2) \leq C_1 - C_2 \int_s^{t+1} r^{-x-y} dr.$$

For (B.7), we find that

$$\|\Xi_t^*\|_F \leq |\sigma_t^2 - \sigma_{t+1}^2|N\|Y X^\mathsf{T}[(X X^\mathsf{T})^+]^2\|_F = O(t^{-y-1}).$$

Finally, for (B.8), we find that

$$\eta_t^2 \mathrm{tr}\left[ \mathrm{Id} \circ \mathrm{Var}\Big( \mathbb{E}[W_t] X_t X_t^\mathsf{T} - Y_t X_t^\mathsf{T} \Big) \right] = \eta_t^2 \mathbb{E}\left[ \|\mathbb{E}[W_t](X_t X_t^\mathsf{T} - \mathbb{E}[X_t X_t^\mathsf{T}]) - (Y_t X_t^\mathsf{T} - \mathbb{E}[Y_t X_t^\mathsf{T}])\|_F^2 \right]$$

$$\leq 2\eta_t^2 \mathbb{E}\left[ \|\mathbb{E}[W_t](X_t X_t^\mathsf{T} - \mathbb{E}[X_t X_t^\mathsf{T}])\|_F^2 + \|Y_t X_t^\mathsf{T} - \mathbb{E}[Y_t X_t^\mathsf{T}]\|_F^2 \right]$$

$$\leq 2\eta_t^2 \Big( \|\mathbb{E}[W_t]\|_F^2 \mathbb{E}[\|X_t X_t^\mathsf{T} - \mathbb{E}[X_t X_t^\mathsf{T}]\|_F^2] + \|Y_t X_t^\mathsf{T} - \mathbb{E}[Y_t X_t^\mathsf{T}]\|_F^2 \Big)$$

$$= O(t^{-2x-y}).$$

Noting finally that $\|W_t^* - W_{\min}\|_F = O(\sigma_t^2) = O(t^{-y})$, we apply Theorem B.4 with $\alpha = x + y$, $\beta_1 = y + 1$, and $\beta_2 = 2x + y$ to obtain the desired estimates. This concludes the proof of Theorem 4.1. $\qquad \square$

## D.2 Proof of Theorem 4.2 for SGD

We now prove Theorem 4.2 for SGD. As before, we will apply Theorems 3.1 and B.4. To check the hypotheses of these theorems, we will use the following expressions for moments of the augmented data matrices.

**Lemma D.1.** *We have*

$$\mathbb{E}[Y_t X_t^\mathsf{T}] = Y X^\mathsf{T} \qquad and \qquad \mathbb{E}[X_t X_t^\mathsf{T}] = X X^\mathsf{T} + \sigma_t^2 N \, \mathrm{Id}_{n\times n}. \tag{D.2}$$

*Moreover,*

$$\mathbb{E}[Y_t X_t^\mathsf{T} X_t Y_t^\mathsf{T}] = c_t^4 \mathbb{E}[Y A_t A_t^\mathsf{T} X^\mathsf{T} X A_t A_t^\mathsf{T} Y^\mathsf{T} + \sigma_t^2 Y A_t G_t^\mathsf{T} G_t A_t^\mathsf{T} Y^\mathsf{T}]$$

$$= \frac{N}{B_t} Y \operatorname{diag}(X^\mathsf{T} X) Y^\mathsf{T} + \frac{B_t - 1}{B_t} Y X^\mathsf{T} X Y^\mathsf{T} + \sigma_t^2 N Y Y^\mathsf{T}$$

$$\mathbb{E}[Y_t X_t^\mathsf{T} X_t X_t^\mathsf{T}] = c_t^4 \mathbb{E}[Y A_t A_t^\mathsf{T} X^\mathsf{T} X A_t A_t^\mathsf{T} X^\mathsf{T} + \sigma_t^2 Y A_t G_t^\mathsf{T} G_t A_t^\mathsf{T} X^\mathsf{T}$$

$$+ \sigma_t^2 Y A_t G_t^\mathsf{T} X A_t G_t^\mathsf{T} + \sigma_t^2 Y A_t A_t^\mathsf{T} X^\mathsf{T} G_t G_t^\mathsf{T}]$$

$$= \frac{N}{B_t} Y \operatorname{diag}(X^\mathsf{T} X) X^\mathsf{T} + \frac{B_t - 1}{B_t} Y X^\mathsf{T} X X^\mathsf{T} + \sigma_t^2 (N + \frac{n+1}{B_t/N}) Y X^\mathsf{T}$$

$$\mathbb{E}[X_t X_t^\mathsf{T} X_t X_t^\mathsf{T}] = c_t^4 \mathbb{E}[X A_t A_t^\mathsf{T} X^\mathsf{T} X A_t A_t^\mathsf{T} X^\mathsf{T} + \sigma_t^2 G_t G_t^\mathsf{T} X A_t A_t^\mathsf{T} X^\mathsf{T} + \sigma_t^2 X A_t G_t^\mathsf{T} G_t A_t^\mathsf{T} X^\mathsf{T}$$

$$+ \sigma_t^2 X A_t A_t^\mathsf{T} X^\mathsf{T} G_t G_t^\mathsf{T} + \sigma_t^2 G_t A_t^\mathsf{T} X^\mathsf{T} G_t A_t^\mathsf{T} X^\mathsf{T} + \sigma_t^2 X A_t G_t^\mathsf{T} X A_t G_t^\mathsf{T}$$

$$+ \sigma_t^2 G_t A_t^\mathsf{T} X^\mathsf{T} X A_t G_t^\mathsf{T} + \sigma_t^4 G_t G_t^\mathsf{T} G_t G_t^\mathsf{T}]$$

$$= \frac{N}{B_t} X \operatorname{diag}(X^\mathsf{T} X) X^\mathsf{T} + \frac{B_t - 1}{B_t} X X^\mathsf{T} X X^\mathsf{T} + \sigma_t^2 (2N + \frac{n+2}{B_t/N}) X X^\mathsf{T}$$

$$+ \sigma_t^2 \frac{N}{B_t} tr(X X^\mathsf{T}) \operatorname{Id}_{n \times n} + \sigma_t^4 N (N + \frac{n+1}{B_t/N}) \operatorname{Id}_{n \times n}.$$

*Proof.* All these formulas are obtained by direct, if slightly tedious, computation. $\qquad\square$

With these expressions in hand, we now check the conditions of Theorem 3.1. First, by the Sherman-Morrison-Woodbury matrix inversion formula, we have

$$\|\Xi_t^*\|_F = |\sigma_t^2 N - \sigma_{t+1}^2 N| \left\| Y X^\mathsf{T} (X X^\mathsf{T} + \sigma_t^2 N \cdot \operatorname{Id}_{n \times n})^{-1} (X X^\mathsf{T} + \sigma_{t+1}^2 N \cdot \operatorname{Id}_{n \times n})^{-1} \right\|_F \tag{D.3}$$

$$\leq N |\sigma_t^2 - \sigma_{t+1}^2| \left\| Y X^\mathsf{T} [(X X^\mathsf{T})^+]^2 \right\|_F.$$

Because $\sigma_t^2$ is non-increasing, this implies (3.7). Next, by Lemma D.1 we have that

$$\sum_{t=0}^\infty \eta_t \lambda_{\min, V_\parallel} (\mathbb{E}[X_t X_t^\mathsf{T}]) \geq \sum_{t=0}^\infty \eta_t \sigma_t^2 = \infty$$

by the first given condition in (4.7), which verifies (3.8). Finally, by Lemma D.1, we may compute

$$\mathbb{E}\left[ \left\| X_t X_t^\mathsf{T} - \mathbb{E}\left[ X_t X_t^\mathsf{T} \right] \right\|_F^2 \right]$$

$$= \frac{1}{B_t} \operatorname{tr}\left( X(N \operatorname{diag}(X^\mathsf{T} X) - X^\mathsf{T} X) X^\mathsf{T} \right) + 2\sigma_t^2 \frac{n+1}{B_t/N} \operatorname{tr}(X X^\mathsf{T}) + \sigma_t^4 \frac{N n(n+1)}{B_t/N}$$

and

$$\mathbb{E}\left[ \|Y_t X_t^\mathsf{T} - \mathbb{E}[Y_t X_t^\mathsf{T}]\|_F^2 \right] = \frac{1}{B_t} \operatorname{tr}\left( Y(N \operatorname{diag}(X^\mathsf{T} X) - X^\mathsf{T} X) Y^\mathsf{T} \right) + \sigma_t^2 N \operatorname{tr}(Y Y^\mathsf{T}).$$

Together, these imply that for some constant $C > 0$, we have that

$$\sum_{t=0}^\infty \eta_t^2 \left[ \mathbb{E}\left[ \left\| X_t X_t^\mathsf{T} - \mathbb{E}\left[ X_t X_t^\mathsf{T} \right] \right\|_F^2 \right] + \mathbb{E}\left[ \left\| Y_t X_t^\mathsf{T} - \mathbb{E}[Y_t X_t^\mathsf{T}] \right\|_F^2 \right] \right] \leq \sum_{t=0}^\infty C \eta_t^2 < \infty$$

by the second given condition in (4.7), which verifies (3.9). Thus the conditions of Theorem 3.1 apply, which shows that $W_t \xrightarrow{p} W_{\min}$, as desired.

For rates of convergence, we check the conditions of Theorem B.4. For (B.6), we will apply Theorem A.6 to bound $\log \mathbb{E} \left\| \prod_{r=s}^t \left( \operatorname{Id} - \frac{2\eta_r}{N} X_r X_r^\mathsf{T} \right) \right\|_2^2$. By the second moment expression $\mathbb{E}[X_r X_r^\mathsf{T}] = X X^\mathsf{T} + \sigma_r^2 N \operatorname{Id}_{n \times n}$, we find that

$$\left\| \mathbb{E}\left[ \operatorname{Id} - \frac{2\eta_r}{N} X_r X_r^\mathsf{T} \right] \right\|_2 = 1 - 2\eta_r \sigma_r^2.$$

Moreover, by Lemma D.1, for some constant $C > 0$ we have

$$\mathbb{E}\left[\left\|\mathrm{Id} - \frac{2\eta_r}{N}X_rX_r^\mathsf{T} - \mathbb{E}\left[\mathrm{Id} - \frac{2\eta_r}{N}X_rX_r^\mathsf{T}\right]\right\|_2^2\right]$$

$$= \frac{4\eta_r^2}{N^2}\mathbb{E}\left[\left\|X_rX_r^\mathsf{T} - \mathbb{E}\left[X_rX_r^\mathsf{T}\right]\right\|_2^2\right]$$

$$\leq \frac{4\eta_r^2}{N^2}\left[\frac{1}{B_t}\mathrm{tr}\left(X(N\,\mathrm{diag}(X^\mathsf{T}X) - X^\mathsf{T}X)X^\mathsf{T}\right) + 2\sigma_t^2\frac{n+1}{B_t/N}\mathrm{tr}(XX^\mathsf{T}) + \sigma_t^4\frac{Nn(n+1)}{B_t/N}\right]$$

$$\leq C\eta_r^2.$$

Applying Theorem A.6 with $a_r = 1 - 2\eta_r\sigma_r^2$ and $b_r^2 = \frac{C\eta_r^2}{a_r^2}$, we find that

$$\log\mathbb{E}\left\|\prod_{r=s}^t\left(\mathrm{Id} - \frac{2\eta_r}{N}X_rX_r^\mathsf{T}\right)\right\|_2^2 \leq \sum_{r=s}^t\frac{C\eta_r^2}{a_r^2} + 2\log\left(\prod_{r=s}^t(1 - 2\eta_r\sigma_r^2)\right)$$

$$\leq \sum_{r=s}^t\frac{C\eta_r^2}{a_r^2} - 4\sum_{r=s}^t\eta_r\sigma_r^2.$$

Because $\eta_r = \Theta(r^{-x})$ and $\sigma_r^2 = \Theta(r^{-y})$, we obtain (B.6) with $\alpha = x + y$, $C_1 = \sum_{r=0}^\infty\frac{C\eta_r^2}{a_r^2}$, and some $C_2 > 0$, where $C_1$ is finite because $x > \frac{1}{2}$. Next, (B.7) holds for $\beta_1 = -y - 1$ by (D.3). Finally, it remains to bound

$$\eta_t^2\mathrm{tr}\left[\mathrm{Id}\circ\mathrm{Var}(\mathbb{E}[W_t]X_tX_t^\mathsf{T} - Y_tX_t^\mathsf{T})\right]$$

to verify (B.8). Again using D.1 and noting that $W_{\min}X = Y$, we find

$$\eta_t^2\mathrm{tr}\left[\mathrm{Id}\circ\mathrm{Var}(\mathbb{E}[W_t]X_tX_t^\mathsf{T} - Y_tX_t^\mathsf{T})\right]$$

$$= \eta_t^2\mathrm{tr}\Big(\frac{1}{B_t}\mathbb{E}[W_t]X(N\,\mathrm{diag}(X^\mathsf{T}X) - X^\mathsf{T}X)X^\mathsf{T}\mathbb{E}[W_t]^\mathsf{T}$$

$$+ 2\sigma_t^2\frac{n+1}{B_t/N}\mathbb{E}[W_t]XX^\mathsf{T}\mathbb{E}[W_t]^\mathsf{T} + (\sigma_t^2\frac{N}{B_t}\mathrm{tr}(XX^\mathsf{T}) + \sigma_t^4 N\frac{n+1}{B_t/N})\mathbb{E}[W_t]\mathbb{E}[W_t]^\mathsf{T}\Big)$$

$$- 2\eta_t^2\mathrm{tr}\Big(\frac{1}{B_t}Y(N\,\mathrm{diag}(X^\mathsf{T}X) - X^\mathsf{T}X)X^\mathsf{T}\mathbb{E}[W_t]^\mathsf{T} + \sigma_t^2\frac{n+1}{B_t/N}YX^\mathsf{T}\mathbb{E}[W_t]^\mathsf{T}\Big)$$

$$+ \eta_t^2\mathrm{tr}\Big(\frac{1}{B_t}Y(N\,\mathrm{diag}(X^\mathsf{T}X) - X^\mathsf{T}X)Y^\mathsf{T} + \sigma_t^2 NYY^\mathsf{T}\Big)$$

$$\leq C\eta_t^2(\sigma_t^2 + \|\Delta_t\|_F^2)$$

for some $C > 0$ and $\Delta_t := \mathbb{E}[W_t] - W_{\min}$. Define also $\Delta_t' := \mathbb{E}[W_t] - W_t^*$ so that, exactly as in Proposition B.6, we have

$$\Delta_{t+1}' = \Delta_t'\left(\mathrm{Id} - \frac{2\eta_t}{N}\mathbb{E}\left[X_tX_t^\mathsf{T}\right]\right) + \frac{2}{N}\Xi_t^*, \qquad \Delta_t' := \mathbb{E}\left[W_t - W_t^*\right].$$

Since $\|\Xi_t^*\|_F = O(t^{-y-1})$ and we already saw that

$$\left\|\mathrm{Id} - \frac{2\eta_t}{N}\mathbb{E}\left[X_tX_t^\mathsf{T}\right]\right\|_2 = 1 - 2\eta_t\sigma_t^2,$$

we may use the single sided decay estimates of Lemma A.4 to conclude that $\|\Delta_t'\|_F = O(t^{x-1})$. This implies that

$$\|\Delta_t\|_F \leq \|\Delta_t'\|_F + \|W_t^* - W_{\min}\|_F = O(t^{x-1}) + \Theta(t^{-y}) = \Theta(t^{-y})$$

since we assumed that $x + y < 1$. Therefore, we obtain

$$\eta_t^2\mathrm{tr}\left[\mathrm{Id}\circ\mathrm{Var}(\mathbb{E}[W_t]X_tX_t^\mathsf{T} - Y_tX_t^\mathsf{T})\right] \leq C\eta_t^2(\sigma_t^2 + \Theta(t^{-2y})) = \Theta(t^{-2x-y}),$$

showing that condition (B.8) holds with $\beta_2 = 2x + y$. We have thus verified all of the conditions of Theorem B.4, whose application completes the proof. $\qquad\square$

# E  Analysis of random projection augmentations

In this section, we give a full analysis of the random projection augmentations presented in Section 5. In this setting, we have a preconditioning matrix $\Pi_t = Q_t \widetilde{\Pi}_t Q_t^\mathsf{T} \in \mathbb{R}^{n \times n}$, where $\widetilde{\Pi}_t$ is a projection matrix and $Q_t$ are Haar random orthogonal matrices. Define the normalized trace of the projection matrix by

$$\gamma_t := \frac{\operatorname{tr}(\Pi_t)}{n}. \tag{E.1}$$

We consider the augmentation given by

$$X_t = \Pi_t X \qquad \text{and} \qquad Y_t = Y.$$

In this setting, we first record the values of the lower order moments of the augmented data matrices, with proofs deferred to Section E.1

**Lemma E.1.** *We have that*

$$\mathbb{E}[X_t] = \gamma_t X \tag{E.2}$$

$$\mathbb{E}[Y_t X_t^\mathsf{T}] = \gamma_t Y X^\mathsf{T} \tag{E.3}$$

$$\mathbb{E}[X_t X_t^\mathsf{T}] = \frac{\gamma_t(\gamma_t + 1/n - 2/n^2)}{1 + 1/n - 2/n^2} X X^\mathsf{T} + \frac{\gamma_t(1 - \gamma_t)}{n(1 + 1/n - 2/n^2)} \|X\|_F^2 \operatorname{Id} \tag{E.4}$$

$$\mathbb{E}[X_t X_t^\mathsf{T} X_t] = \frac{\gamma_t^2 + \gamma_t(1/n - 2/n^2)}{1 + 1/n - 2/n^2} X X^\mathsf{T} X + \frac{\gamma_t(1 - \gamma_t)}{1 + 1/n - 2/n^2} \frac{\|X\|_F^2}{n} X \tag{E.5}$$

$$\mathbb{E}[\|X_t X_t^\mathsf{T}\|_F^2] = \frac{(\gamma_t^2 + \gamma_t(1/n - 2/n^2)) \|X X^\mathsf{T}\|_F^2 + \gamma_t(1 - \gamma_t) \frac{1}{n} \|X\|_F^4}{1 + 1/n - 2/n^2}. \tag{E.6}$$

We now compute the proxy loss and its optima as follows.

**Lemma E.2.** *The proxy loss for randomly rotated projections with normalized trace $\gamma_t$ given by (E.1) is*

$$\overline{\mathcal{L}}_t(W) = \frac{1}{N} \|Y - \gamma_t W X\|_F^2 + \frac{1}{N} \gamma_t(1 - \gamma_t) \|X\|_F^2 \|W\|_F^2 + \frac{\gamma_t(1 - \gamma_t)}{N} \frac{1/n - 2/n^2}{1 + 1/n - 2/n^2} (\|WX\|_F^2 + \frac{1}{n} \|X\|_F^2 \|W\|_F^2).$$

*It has proxy optima*

$$W_t^* = Y X^\mathsf{T} \Big( \frac{\gamma_t + 1/n - 2/n^2}{1 + 1/n - 2/n^2} X X^\mathsf{T} + \frac{1 - \gamma_t}{1 + 1/n - 2/n^2} \frac{\|X\|_F^2}{n} \operatorname{Id} \Big)^{-1}.$$

*Proof.* Applying (3.1), we find that

$$\overline{\mathcal{L}}_t(W) = \frac{1}{N} \mathbb{E}[\|Y_t - W X_t\|_F^2] = \frac{1}{N} \mathbb{E}[\|Y_t\|_F^2] - \frac{2}{N} \mathbb{E}[\operatorname{tr}(X_t^\mathsf{T} W^\mathsf{T} Y_t)] + \frac{1}{N} \mathbb{E}[\operatorname{tr}(W^\mathsf{T} W X_t X_t^\mathsf{T})].$$

Applying Lemma E.1, we conclude that

$$\overline{\mathcal{L}}_t(W) = \frac{1}{N} \|Y\|_F^2 - \frac{2\gamma_t}{N} \operatorname{tr}(X W^\mathsf{T} Y) + \frac{1}{N} \frac{\gamma_t(\gamma_t + 1/n - 2/n^2)}{1 + 1/n - 2/n^2} \|WX\|_F^2 + \frac{1}{N} \frac{\gamma_t(1 - \gamma_t)}{n(1 + 1/n - 2/n^2)} \|X\|_F^2 \|W\|_F^2$$

$$= \frac{1}{N} \|Y - \gamma_t W X\|_F^2 + \frac{1}{N} \gamma_t(1 - \gamma_t) \|X\|_F^2 \|W\|_F^2 + \frac{\gamma_t(1 - \gamma_t)}{N} \frac{1/n - 2/n^2}{1 + 1/n - 2/n^2} (\|WX\|_F^2 + \frac{1}{n} \|X\|_F^2 \|W\|_F^2).$$

The formula for $W_t^*$ then results from Lemma E.1 and the general formula (3.4). $\square$

## E.1  Proof of Lemma E.1

We apply the Weingarten calculus [7] to compute integrals of polynomial functions of the matrix entries of a Haar orthogonal matrix. Because each matrix entry of the expectations in Lemma E.1 is such a polynomial, this will allow us to compute the relevant expectations. The main result we use is the following on polynomials of degree at most $4$ and its corollary.

**Proposition E.3** (Corollary 3.4 of [7]). *For an $n \times n$ orthogonal matrix $Q$ drawn from the Haar measure, we have that*

$$\mathbb{E}[Q_{i_1 j_1} Q_{i_2 j_2}] = \frac{1}{n} \delta_{i_1 i_2} \delta_{j_1 j_2} \tag{E.7}$$

$$\mathbb{E}[Q_{i_1 j_1} Q_{i_2 j_2} Q_{i_3 j_3} Q_{i_4 j_4}] = \frac{-1}{n(n-1)(n+2)} \Big( \delta_{i_1 i_2} \delta_{i_3 i_4} + \delta_{i_1 i_3} \delta_{i_2 i_4} + \delta_{i_1 i_4} \delta_{i_2 i_3} \Big) \Big( \delta_{j_1 j_2} \delta_{j_3 j_4} + \delta_{j_1 j_3} \delta_{j_2 j_4} + \delta_{j_1 j_4} \delta_{j_2 j_3} \Big)$$
$$+ \frac{1}{n(n-1)} \Big( \delta_{i_1 i_2} \delta_{i_3 i_4} \delta_{j_1 j_2} \delta_{j_3 j_4} + \delta_{i_1 i_3} \delta_{i_2 i_4} \delta_{j_1 j_3} \delta_{j_2 j_4} + \delta_{i_1 i_4} \delta_{i_2 i_3} \delta_{j_1 j_4} \delta_{j_2 j_3} \Big), \tag{E.8}$$

*where $\delta_{ab}$ denotes the Kronecker delta function $\delta_{ab} = \mathbf{1}\{a = b\}$.*

**Corollary E.4.** *For matrices $A, B, C \in \mathbb{R}^{n \times n}$ and an $n \times n$ Haar random orthogonal matrix $Q$, we have*

$$\mathbb{E}[QAQ^{\mathsf{T}}] = \frac{1}{n} tr(A) \cdot \mathrm{Id} \tag{E.9}$$

$$\mathbb{E}[QAQ^{\mathsf{T}} BQCQ^{\mathsf{T}}]_{ij} = -\frac{1}{n(n-1)(n+2)} \Big( tr(A)tr(C) + tr(AC^{\mathsf{T}} + AC) \Big) \Big( \delta_{ij} tr(B) + B_{ij} + B_{ji} \Big)$$
$$+ \frac{1}{n(n-1)} \Big( tr(A)tr(C)B_{ij} + tr(AC^{\mathsf{T}})B_{ji} + \delta_{ij} tr(AC)tr(B) \Big). \tag{E.10}$$

*Proof.* For (E.9), notice by (E.7) that

$$\mathbb{E}[QAQ^{\mathsf{T}}]_{ij} = \sum_{a,b=1}^{n} \mathbb{E}[Q_{ia} A_{ab} Q_{jb}] = \sum_{a,b=1}^{n} \delta_{ab} \delta_{ij} \frac{1}{n} A_{ab} = \delta_{ij} \frac{1}{n} tr(A).$$

For (E.10), notice by (E.8) that

$$\mathbb{E}[QAQ^{\mathsf{T}} BQCQ^{\mathsf{T}}]_{ij} = \sum_{a,b,c,d,e,f=1}^{n} \mathbb{E}[Q_{ia} A_{ab} Q_{cb} B_{cd} Q_{de} C_{ef} Q_{jf}]$$

$$= -\frac{1}{n(n-1)(n+2)} \Big( \sum_{a,e=1}^{n} A_{aa} C_{ee} + \sum_{a,b=1}^{n} A_{ab} C_{ab} + \sum_{a,b=1}^{n} A_{ab} C_{ba} \Big) \Big( \sum_{c=1}^{n} \delta_{ij} B_{cc} + B_{ij} + B_{ji} \Big)$$

$$+ \frac{1}{n(n-1)} \Big( \sum_{a,e=1}^{n} A_{aa} C_{ee} B_{ij} + \sum_{a,b=1}^{n} A_{ab} C_{ab} B_{ji} + \delta_{ij} \sum_{a,b=1}^{n} A_{ab} C_{ba} \sum_{c=1}^{n} B_{cc} \Big),$$

which when simplified gives the desired conclusion. $\qquad\square$

We now compute each lower order moment; in each computation, let $Q$ denote a random $n \times n$ orthogonal matrix drawn from the Haar measure. Claims (E.2) and (E.3) follow from Corollary E.4 and the facts that

$$\mathbb{E}[X_t] = \mathbb{E}[Q\widetilde{\Pi}_t Q^{\mathsf{T}}]X \qquad \text{and} \qquad \mathbb{E}[Y_t X_t^{\mathsf{T}}] = Y \mathbb{E}[Q\widetilde{\Pi}_t Q^{\mathsf{T}}]X.$$

Claims (E.4) and (E.5) follow from Corollary E.4 and the facts that

$$\mathbb{E}[X_t X_t^{\mathsf{T}}] = \mathbb{E}[Q\widetilde{\Pi}_t Q^{\mathsf{T}} X X^{\mathsf{T}} Q\widetilde{\Pi}_t Q^{\mathsf{T}}] \qquad \text{and} \qquad \mathbb{E}[X_t X_t^{\mathsf{T}} X_t] = \mathbb{E}[Q\widetilde{\Pi}_t Q^{\mathsf{T}} X X^{\mathsf{T}} Q\widetilde{\Pi}_t^2 Q^{\mathsf{T}}]X.$$

Finally, (E.6) follows from Corollary E.4 and the fact that

$$\mathbb{E}[\|X_t X_t^{\mathsf{T}}\|_F^2] = \mathbb{E}[\mathrm{tr}(Q\widetilde{\Pi}_t Q^{\mathsf{T}} X X^{\mathsf{T}} Q\widetilde{\Pi}_t^2 Q^{\mathsf{T}} X X^{\mathsf{T}} Q\widetilde{\Pi}_t Q^{\mathsf{T}})] = \mathbb{E}[\mathrm{tr}(\widetilde{\Pi}_t Q^{\mathsf{T}} X X^{\mathsf{T}} Q\widetilde{\Pi}_t^2 Q^{\mathsf{T}} X X^{\mathsf{T}} Q\widetilde{\Pi}_t))].$$

This completes the proof of Lemma E.1.

## E.2 Proof of Theorem 5.1

We first show convergence. By Lemma E.1, we find that $V_\parallel = \mathbb{R}^n$. Furthermore, because $\gamma_t \to 1$, we have that $W_\infty^* = \lim_{t\to\infty} W_t^* = W_{\min}$. It therefore suffices to verify the conditions of Theorem 3.1. First, notice that

$$
\begin{aligned}
\|\Xi_t^*\|_F &= \frac{|\gamma_t - \gamma_{t+1}|}{1 + 1/n - 2/n^2} \left\| YX^\mathsf{T} \Big( \frac{\gamma_{t+1} + 1/n - 2/n^2}{1 + 1/n - 2/n^2} XX^\mathsf{T} + \frac{1 - \gamma_{t+1}}{1 + 1/n - 2/n^2} \frac{\|X\|_F^2}{n} \operatorname{Id} \Big)^{-1} \right. \\
&\qquad \left. \Big( XX^\mathsf{T} - \frac{\|X\|_F^2}{n} \operatorname{Id} \Big) \Big( \frac{\gamma_t + 1/n - 2/n^2}{1 + 1/n - 2/n^2} XX^\mathsf{T} + \frac{1 - \gamma_t}{1 + 1/n - 2/n^2} \frac{\|X\|_F^2}{n} \operatorname{Id} \Big)^{-1} \right\|_F \\
&\le \frac{|\gamma_t^{-1} - \gamma_{t+1}^{-1}|}{1 + 1/n - 2/n^2} \|YX^\mathsf{T}\|_F \|[XX^\mathsf{T}]^+\|_F^2 \Big\| XX^\mathsf{T} - \frac{1}{n}\|X\|_F^2 \operatorname{Id} \Big\|_F.
\end{aligned}
$$

Because $\gamma_t$ are increasing and $\lim_{t\to\infty} \gamma_t = 1$, we find that (3.7) holds. Now, by Lemma E.1, we have

$$
\lambda_{\min}(\mathbb{E}[X_t X_t^\mathsf{T}]) \ge \frac{\gamma_t(1 - \gamma_t)}{1 + 1/n - 2/n^2} \frac{\|X\|_F^2}{n},
$$

so the first condition in (5.2) implies (3.8). Finally, using Lemma E.1 we may compute

$$
\begin{aligned}
\mathbb{E}[\|X_t X_t^\mathsf{T} - \mathbb{E}[X_t X_t^\mathsf{T}]\|_F^2] &= \mathbb{E}[\|X_t X_t\|_F^2] - \|\mathbb{E}[X_t X_t^\mathsf{T}]\|_F^2 \\
&= \frac{\gamma_t(1 - \gamma_t)(\gamma_t(1 + \gamma_t)n^4 + (1 + 2\gamma_t)n^3 - (1 + 4\gamma_t)n^2 - 4n + 4)}{(n-1)^2(n+2)^2} \|XX^\mathsf{T}\|_F^2 \\
&\quad + \frac{\gamma_t(1 - \gamma_t)((1 - \gamma_t - \gamma_t^2)n^4 + (1 - 2\gamma_t)n^3 + (4\gamma_t - 2)n^2)}{(n-1)^2(n+2)^2} \frac{1}{n}\|X\|_F^4 \\
&\le 2\gamma_t(1 - \gamma_t)\Big( \|XX^\mathsf{T}\|_F^2 + \frac{1}{n}\|X\|_F^4 \Big) \tag{E.11}
\end{aligned}
$$

and also

$$
\begin{aligned}
\mathbb{E}[\|Y_t X_t^\mathsf{T} - \mathbb{E}[Y_t X_t^\mathsf{T}]\|_F^2] &= \mathbb{E}[\|Y_t X_t^\mathsf{T}\|_F^2] - \|\mathbb{E}[Y_t X_t^\mathsf{T}]\|_F^2 \\
&= \frac{\gamma_t(1 - \gamma_t)(1/n - 2/n^2)}{1 + 1/n - 2/n^2} \|YX^\mathsf{T}\|_F^2 + \frac{\gamma_t(1 - \gamma_t)}{1 + 1/n - 2/n^2} \frac{1}{n}\|X\|_F^2\|Y\|_F^2 \\
&\le \gamma_t(1 - \gamma_t)\Big( \|YX^\mathsf{T}\|_F^2 + \frac{1}{n}\|X\|_F^2\|Y\|_F^2 \Big). \tag{E.12}
\end{aligned}
$$

Combining these bounds with the second condition in (5.2) gives (3.9). Having verified (3.7), (3.8), and (3.9), we conclude by Theorem 3.1 that $W_t \xrightarrow{p} W_{\min}$ as desired.

We now obtain rates using Theorem B.4. We aim to show that if $\eta_t = \Theta(t^{-x})$ and $\gamma_t = 1 - \Theta(t^{-y})$ with $x, y > 0$, $x + y < 1$, and $2x + y > 1$, then for any $\varepsilon > 0$ we have that

$$
t^{\min\{y, \frac{1}{2}x\} - \varepsilon} \|W_t - W_{\min}\|_F \xrightarrow{p} 0.
$$

We now check the hypotheses of Theorem B.4. For (B.6), by Lemma E.1 and (E.11), $Y_r = \operatorname{Id} - \frac{2\eta_r}{N} X_r X_r^\mathsf{T}$ satisfies the hypotheses of Theorem A.6 with

$$
a_r = 1 - \frac{2\eta_r}{N} \frac{\gamma_t(1 - \gamma_t)}{1 + 1/n - 2/n^2} \frac{\|X\|_F^2}{n} \qquad \text{and} \qquad b_r^2 = \frac{8\eta_r^2 \gamma_t(1 - \gamma_t)}{a_r^2 N^2}\Big( \|XX^\mathsf{T}\|_F^2 + \frac{1}{n}\|X\|_F^4 \Big).
$$

By Theorem A.6 and the fact that $\eta_t = \Theta(t^{-x})$ and $\gamma_t = 1 - \Theta(t^{-y})$, we find for some $C_1, C_2 > 0$ that

$$
\begin{aligned}
\log \mathbb{E}\left[ \left\| \prod_{r=s}^t \Big( \operatorname{Id} - \frac{2\eta_r}{N} X_r X_r^\mathsf{T} \Big) \right\|_2^2 \right] &\le \sum_{r=s}^t b_r^2 + 2\sum_{r=s}^t \log\Big(1 - \frac{2\eta_r}{N(1 + 1/n - 2/n^2)} \gamma_t(1 - \gamma_t)\frac{\|X\|_F^2}{n}\Big) \\
&\le C_1 - C_2 \int_s^{t+1} r^{-x-y} dr.
\end{aligned}
$$

For (B.7), our previous computations show that

$$\|\Xi_t^*\|_F \leq \frac{|\gamma_t^{-1} - \gamma_{t+1}^{-1}|}{1 + 1/n - 2/n^2} \|YX^\mathsf{T}\|_F \|[XX^\mathsf{T}]^+\|_F^2 \|XX^\mathsf{T} - \frac{1}{n}\|X\|_F^2 \,\mathrm{Id}\,\|_F = O(t^{-y-1}).$$

Finally, for (B.8), we find by (E.11), (E.12) and the fact that $\|\mathbb{E}[W_t]\|_F = O(1)$ that

$$\eta_t^2 \mathrm{tr}\left[\mathrm{Id} \circ \mathrm{Var}\left(\mathbb{E}[W_t]X_t X_t^\mathsf{T} - Y_t X_t^\mathsf{T}\right)\right] = \eta_t^2 \mathbb{E}\left[\|\mathbb{E}[W_t](X_t X_t^\mathsf{T} - \mathbb{E}[X_t X_t^\mathsf{T}]) - (Y_t X_t^\mathsf{T} - \mathbb{E}[Y_t X_t^\mathsf{T}])\|_F^2\right]$$

$$\leq 2\eta_t^2 \mathbb{E}\left[\|\mathbb{E}[W_t](X_t X_t^\mathsf{T} - \mathbb{E}[X_t X_t^\mathsf{T}])\|_F^2 + \|Y_t X_t^\mathsf{T} - \mathbb{E}[Y_t X_t^\mathsf{T}]\|_F^2\right]$$

$$\leq 2\eta_t^2 \left(\|\mathbb{E}[W_t]\|_F^2 \mathbb{E}[\|X_t X_t^\mathsf{T} - \mathbb{E}[X_t X_t^\mathsf{T}]\|_F^2] + \|Y_t X_t^\mathsf{T} - \mathbb{E}[Y_t X_t^\mathsf{T}]\|_F^2\right)$$

$$= O(t^{-2x-y}).$$

Noting finally that $\|W_t^* - W_{\min}\|_F = O(1 - \gamma_t) = O(t^{-y})$, we apply Theorem B.4 with $\alpha = x + y$, $\beta_1 = y + 1$, and $\beta_2 = 2x + y$ to obtain the desired rate. This concludes the proof of Theorem 5.1. $\square$

### E.3 Experimental validation

To validate Theorem 5.1, we ran augmented GD with random projection augmentation on $N = 100$ simulated datapoints. Inputs were i.i.d. Gaussian vectors in dimension $n = 400$, and outputs in dim $p = 1$ were generated by a random linear map with i.i.d Gaussian coefficients drawn from $\mathcal{N}(1, 1)$. The learning rate followed a fixed polynomially decaying schedule $\eta_t = \frac{0.005}{100} \cdot (\text{batch size}) \cdot (1 + \frac{t}{20})^{-0.66}$. Figure E.1 shows MSE and $\|W_{t,\perp}\|_F$ along a single optimization trajectory with different schedules for the dimension ratio $1 - \gamma_t$ used in random projection augmentation. Code to generate this figure is provided in `supplement.zip` in the supplement. It ran in 30 minutes on a standard laptop CPU.

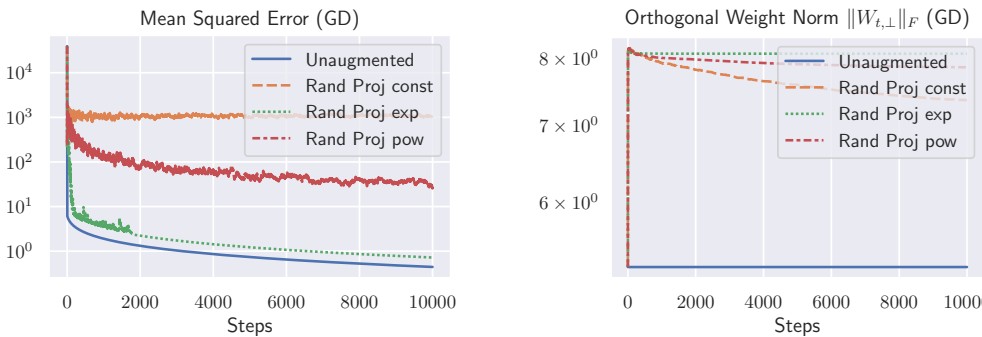

Figure E.1: MSE and $\|W_{t,\perp}\|_F$ for optimization trajectories of GD with random projection augmentation Jointly scheduling learning rate and noise variance to have polynomial decay is necessary for optimization to converge to the minimal norm solution $W_{\min}$. Rand Proj const, Rand Proj exp, and Rand Proj pow have random projection augmentation with $1 - \gamma_t = 0.1, 0.1e^{-0.02t}, 0.1(1 + \frac{t}{50})^{-0.33}$, respectively. Other experimental details are the same as those described in §E.3.

## F  Analysis of SGD

This section gives an analysis of vanilla SGD using our method. Let us briefly recall the notation. As before, we consider overparameterized linear regression with loss

$$\mathcal{L}(W; \mathcal{D}) = \frac{1}{N}\|WX - Y\|_F^2,$$

where the dataset $\mathcal{D}$ of size $N$ consists of data matrices $X, Y$ that each have $N$ columns $x_i \in \mathbb{R}^n, y_i \in \mathbb{R}^p$ with $n > N$. We optimize $\mathcal{L}(W; \mathcal{D})$ by augmented SGD either with or without additive Gaussian noise. In the former case, this means that at each time $t$ we replace $\mathcal{D} = (X, Y)$ by a random batch $\mathcal{B}_t = (X_t, Y_t)$ given by a prescribed batch size $B_t = |\mathcal{B}_t|$ in which each datapoint in $\mathcal{B}_t$ is chosen

uniformly with replacement from $\mathcal{D}$, and the resulting data matrices $X_t$ and $Y_t$ are scaled so that $\overline{\mathcal{L}}_t(W) = \mathcal{L}(W; \mathcal{D})$. Concretely, this means that for the normalizing factor $c_t := \sqrt{N/B_t}$ we have

$$X_t = c_t X A_t \qquad \text{and} \qquad Y_t = c_t Y A_t, \tag{F.1}$$

where $A_t \in \mathbb{R}^{N \times B_t}$ has i.i.d. columns $A_{t,i}$ with a single non-zero entry equal to 1 chosen uniformly at random. In this setting the minimum norm optimum for each $t$ are the same and given by

$$W_t^* = W_{\min} = Y X^\mathsf{T} (X X^\mathsf{T})^+,$$

which coincides with the minimum norm optimum for the unaugmented loss.

**Theorem F.1.** *If the learning rate satisfies $\eta_t \to 0$ and*

$$\sum_{t=0}^{\infty} \eta_t = \infty, \tag{F.2}$$

*then for any initialization $W_0$, we have $W_t Q_{\parallel} \xrightarrow{p} W_{\min}$. If further we have that $\eta_t = \Theta(t^{-x})$ with $0 < x < 1$, then for some $C > 0$ we have*

$$e^{C t^{1-x}} \|W_t Q_{\parallel} - W_{\min}\|_F \xrightarrow{p} 0.$$

Theorem F.1 recovers the exponential convergence rate for SGD in the overparametrized settting, which has been previously studied through both empirical and theoretical means [22]. Because $1 \leq B_t \leq N$ for all $t$, it does not affect the asymptotic results in Theorem F.1. In practice, however, the number of optimization steps $t$ is often small enough that $\frac{B_t}{N}$ is of order $t^{-\alpha}$ for some $\alpha > 0$, meaning the choice of $B_t$ can affect rates in this non-asymptotic regime. Though we do not attempt to push our generic analysis to this granularity, this is done in [22] to derive optimal batch sizes and learning rates in the overparametrized setting.

*Proof of Theorem F.1.* In order to apply Theorems B.1 and B.4, we begin by computing the moments of $A_t$ as follows. Recall the notation $\mathrm{diag}(M)$ from Appendix A.1.

**Lemma F.2.** *For any $Z \in \mathbb{R}^{N \times N}$, we have that*

$$\mathbb{E}[A_t A_t^\mathsf{T}] = \frac{B_t}{N} \mathrm{Id}_{N \times N} \qquad \text{and} \qquad \mathbb{E}[A_t A_t^\mathsf{T} Z A_t A_t^\mathsf{T}] = \frac{B_t}{N} \mathrm{diag}(Z) + \frac{B_t(B_t - 1)}{N^2} Z.$$

*Proof.* We have that

$$\mathbb{E}[A_t A_t^\mathsf{T}] = \sum_{i=1}^{B_t} \mathbb{E}[A_{i,t} A_{i,t}^\mathsf{T}] = \frac{B_t}{N} \mathrm{Id}_{N \times N} .$$

Similarly, we find that

$$
\begin{aligned}
\mathbb{E}[A_t A_t^\mathsf{T} Z A_t A_t^\mathsf{T}] &= \sum_{i,j=1}^{B_t} \mathbb{E}[A_{i,t} A_{i,t}^\mathsf{T} Z A_{j,t} A_{j,t}^\mathsf{T}] \\
&= \sum_{i=1}^{B_t} \mathbb{E}[A_{i,t} A_{i,t}^\mathsf{T} Z A_{i,t} A_{i,t}^\mathsf{T}] + 2 \sum_{1 \leq i < j \leq B_t} \mathbb{E}[A_{i,t} A_{i,t}^\mathsf{T} Z A_{j,t} A_{j,t}^\mathsf{T}] \\
&= \frac{B_t}{N} \mathrm{diag}(Z) + \frac{B_t(B_t - 1)}{N^2} Z,
\end{aligned}
$$

which completes the proof. $\qquad \square$

Let us first check convergence in mean:

$$\mathbb{E}[W_t] Q_{\parallel} \to W_{\min}.$$

To see this, note that Lemma F.2 implies

$$\mathbb{E}[Y_t X_t^\mathsf{T}] = Y X^\mathsf{T} \qquad \mathbb{E}[X_t X_t^\mathsf{T}] = X X^\mathsf{T},$$

which yields that

$$W_t^* = YX^{\mathsf{T}}[XX^{\mathsf{T}}]^+ = W_{\min} \tag{F.3}$$

for all $t$. We now prove convergence. Since all $W_t^*$ are equal to $W_{\min}$, we find that $\Xi_t^* = 0$. By (B.9) and Lemma F.2 we have

$$\mathbb{E}[W_{t+1}] - W_{\min} = (\mathbb{E}[W_t] - W_{\min})\Big(\mathrm{Id} - \frac{2\eta_t}{N}XX^{\mathsf{T}}\Big),$$

which implies since $\frac{2\eta_t}{N} < \lambda_{\max}(XX^{\mathsf{T}})^{-1}$ for large $t$ that for some $C > 0$ we have

$$\|\mathbb{E}[W_t]Q_\parallel - W_{\min}\|_F \le \|W_0 Q_\parallel - W_{\min}\|_F \prod_{s=0}^{t-1}\Big\|Q_\parallel - \frac{2\eta_s}{N}XX^{\mathsf{T}}\Big\|_2$$

$$\le C\|W_0 Q_\parallel - W_{\min}\|_F \exp\Big(-\sum_{s=0}^{t-1}\frac{2\eta_s}{N}\lambda_{\min,V_\parallel}(XX^{\mathsf{T}})\Big). \tag{F.4}$$

From this we readily conclude using (F.2) the desired convergence in mean $\mathbb{E}[W_t]Q_\parallel \to W_{\min}$.

Let us now prove that the variance tends to zero. By Proposition B.6, we find that $Z_t = \mathbb{E}[(W_t - \mathbb{E}[W_t])^{\mathsf{T}}(W_t - \mathbb{E}[W_t])]$ has two-sided decay of type $(\{A_t\}, \{C_t\})$ with

$$A_t = \frac{2\eta_t}{N}X_t X_t^{\mathsf{T}}, \qquad C_t = \frac{4\eta_t^2}{N^2}\big[\mathrm{Id}\circ\mathrm{Var}((\mathbb{E}[W_t]X_t - Y_t)X_t^{\mathsf{T}})\big].$$

To understand the resulting rating of convergence, let us first obtain a bound on $\mathrm{tr}(C_t)$. To do this, note that for any matrix $A$, we have

$$\mathrm{tr}\,(\mathrm{Id}\circ\mathrm{Var}[A]) = \mathrm{tr}\,\Big(\mathbb{E}\big[A^{\mathsf{T}}A\big] - \mathbb{E}\,[A]^{\mathsf{T}}\,\mathbb{E}\,[A]\Big).$$

Moreover, using the definition (F.1) of the matrix $A_t$ and writing

$$M_t := \mathbb{E}\,[W_t]\,X - Y,$$

we find

$$\big((\mathbb{E}\,[W_t]\,X_t - Y_t)X_t^{\mathsf{T}}\big)^{\mathsf{T}}\,(\mathbb{E}\,[W_t]\,X_t - Y_t)X_t^{\mathsf{T}} = XA_t A_t^{\mathsf{T}}M_t^{\mathsf{T}}M_t A_t A_t^{\mathsf{T}}X^{\mathsf{T}}$$

as well as

$$\mathbb{E}\big[\big((\mathbb{E}[W_t]X_t - Y_t)X_t^{\mathsf{T}}\big)\big]^{\mathsf{T}}\,\mathbb{E}\big[(\mathbb{E}[W_t]X_t - Y_t)X_t^{\mathsf{T}}\big] = X\mathbb{E}\big[A_t A_t^{\mathsf{T}}\big]\,M_t^{\mathsf{T}}M_t\mathbb{E}\big[A_t A_t^{\mathsf{T}}\big]\,X^{\mathsf{T}}.$$

Hence, using the expression from Lemma F.2 for the moments of $A_t$ and recalling the scaling factor $c_t = (N/B_t)^{1/2}$, we find

$$\mathrm{tr}(C_t) = \frac{4\eta_t^2}{B_t}\mathrm{tr}\,\Big(X\Big\{\mathrm{diag}\,(M_t^{\mathsf{T}}M_t) - \frac{1}{N}M_t^{\mathsf{T}}M_t\Big\}X^{\mathsf{T}}\Big).$$

Next, writing

$$\Delta_t := \mathbb{E}[W_t] - W_{\min}$$

and recalling (F.3), we see that

$$M_t = \Delta_t X.$$

Thus, applying the estimates (F.4) about exponential convergence of the mean, we obtain

$$\mathrm{tr}(C_t) \le \frac{8\eta_t^2}{B_t}\|\Delta_t Q_\parallel\|_2^2\,\|XX^T\|_2^2 \le C\frac{8\eta_t^2}{B_t}\|XX^T\|_2^2\,\|\Delta_0 Q_\parallel\|_F^2\exp\Big(-\sum_{s=0}^{t-1}\frac{4\eta_s}{N}\lambda_{\min,V_\parallel}(XX^{\mathsf{T}})\Big). \tag{F.5}$$

Notice now that $Y_r = Q_\parallel - A_r$ satisfies the conditions of Theorem A.6 with $a_r = 1 - 2\eta_r\frac{1}{N}\lambda_{\min,V_\parallel}(XX^{\mathsf{T}})$ and $b_r^2 = \frac{4\eta_r^2}{B_r a_r^2 N}\mathrm{tr}\Big(X\,\mathrm{diag}(X^{\mathsf{T}}X)X - \frac{1}{N}XX^{\mathsf{T}}XX^{\mathsf{T}}\Big)$. By Theorem A.6 we then obtain for any $t > s > 0$ that

$$\mathbb{E}\left[\left\|\prod_{r=s+1}^{t}(Q_\parallel - A_r)\right\|_2^2\right] \le e^{\sum_{r=s+1}^{t}b_r^2}\prod_{r=s+1}^{t}\Big(1 - 2\eta_r\frac{1}{N}\lambda_{\min,V_\parallel}(XX^{\mathsf{T}})\Big)^2. \tag{F.6}$$

By two-sided decay of $Z_t$, we find by (F.5), (F.6), and (A.9) that

$$\mathbb{E}[\|W_t Q_\| - \mathbb{E}[W_t]Q_\|\|_F^2] = \text{tr}(Q_\| Z_t Q_\|)$$

$$\leq e^{-\frac{4}{N}\lambda_{\min, V_\|}(XX^\mathsf{T})\sum_{s=0}^{t-1}\eta_s} \frac{\|XX^\mathsf{T}\|_2^2}{N^2}\|\Delta_0 Q_\|\|_F^2 C \sum_{s=0}^{t-1} \frac{8\eta_s^2}{B_s/N} e^{\frac{4\eta_s}{N}\lambda_{\min, V_\|}(XX^\mathsf{T}) + \sum_{r=s+1}^{t} b_r^2}. \quad (\text{F.7})$$

Since $\eta_s \to 0$, we find that $\eta_s \frac{N}{B_s} e^{\frac{4\eta_s}{N}\lambda_{\min, V_\|}(XX^\mathsf{T})}$ is uniformly bounded and that $b_r^2 \leq \frac{4}{N}\lambda_{\min, V_\|}(XX^\mathsf{T})\eta_r$ for sufficiently large $r$. We therefore find that for some $C' > 0$,

$$\mathbb{E}[\|W_t Q_\| - \mathbb{E}[W_t]Q_\|\|_F^2] \leq C' \sum_{s=0}^{t-1} \eta_s e^{-\frac{4}{N}\lambda_{\min, V_\|}(XX^\mathsf{T})\sum_{r=0}^{s}\eta_r},$$

hence $\lim_{t\to\infty} \mathbb{E}[\|W_t Q_\| - \mathbb{E}[W_t]Q_\|\|_F^2] = 0$ by Lemma A.8. Combined with the fact that $\mathbb{E}[W_t]Q_\| \to W_{\min}$, this implies that $W_t Q_\| \xrightarrow{p} W_{\min}$.

To obtain a rate of convergence, observe that by (F.4) and the fact that $\eta_t = \Theta(t^{-x})$, for some $C_1, C_2 > 0$ we have

$$\|\mathbb{E}[W_t]Q_\| - W_{\min}\|_F \leq C_1 \exp\left(-C_2 t^{1-x}\right). \quad (\text{F.8})$$

Similarly, by (F.7) and the fact that $\frac{\eta_s}{B_s/N} < \infty$ uniformly, for some $C_3, C_4, C_5 > 0$ we have

$$\mathbb{E}[\|W_t Q_\| - \mathbb{E}[W_t]Q_\|\|_F^2] \leq C_3 \exp\left(-C_4 t^{1-x}\right)t^{1-x}$$

We conclude by Chebyshev's inequality that for any $a > 0$ we have

$$\mathbb{P}\left(\|W_t Q_\| - W_{\min}\|_F \geq C_1 \exp\left(-C_2 t^{1-x}\right) + a \cdot \sqrt{C_3} t^{\frac{1}{2}-\frac{x}{2}} e^{-C_4 t^{1-x}/2}\right) \leq a^{-2}.$$

Taking $a = t$, we conclude as desired that for some $C > 0$, we have

$$e^{Ct^{1-x}}\|W_t Q_\| - W_{\min}\|_F \xrightarrow{p} 0.$$

This completes the proof of Theorem F.1. $\qquad\square$