# OpenReview forum: "How Data Augmentation affects Optimization for Linear Regression"
_NeurIPS.cc/2021/Conference — NeurIPS 2021 Poster_

### Official Review · Reviewer_sQPi · 2021-07-04

**Rating:** 7
**Confidence:** 4

**Summary:**

This paper provides a theoretical framework on the convergence of (S)GD when considering learning rate and data augmentation. It presents the consistency and convergence rate results of a general optimization method, and gives some detailed examples of adaptive Gaussian noise and random projection.


**Limitations And Societal Impact:**

Some limitations are mentioned in the above.
There is no negative social impact.

**Main Review:**

Originality: this paper is novel based on my knowledge. It provides a new theoretical framework to justify the convergence of optimization under changing learning rate and (either random or arbitrary) data augmentation. The results not only covers the situations where data augmentation is independent to iteration t, but also include more general results when data augmentation is related to t.

Quality and clarity: the paper is well written and the main text is easy to understand. The claims are well supported by the theoretical results. The detailed examples for the general theorems are clearly written. The authors are careful and mention both strengths and weaknesses (only for linear regression model) of this work.

Some minor issue:

(1) the current numerical experiments are weak to support the claims. However, since this paper is already well supported by theories, please consider moving the experiments to appendix and emphasize some other important things.

(2) the statements from line 120 to 128 are somewhat confusing. Please specify why they cannot be obtained from Theorem 3.1/3.2. If A_t is randomly generated and independent to iteration t, they can be viewed as some new data distribution, so they are supposed to converge in the optimization.

Significance: this paper provides detailed theories in the optimization of linear model, extending to other complicated algorithms is challenging, but it could potentially provide deeper understanding in data augmentation.





**Time Spent Reviewing:**

5

---

> ### Author Response · Authors · 2021-08-05
> **Response to review**
>
> We thank the referee for their comments and suggestions, and respond to each below.
>
> **Our results apply to both Mixup and label-preserving transformations.**
>
> We agree that Mixup and label-preserving transformations fall under our framework and can be handled in the way the referee suggests (this was the intended meaning of lines 120-128).  However, in these cases, we have not provided an explicit characterization of the learned minimum beyond the general description in Theorem 3.1 and have thus not discussed them in further detail.  We will be happy to clarify this in the camera-ready version.
>
> **We’ve included experiments because we believe they are helpful for the community.**
>
> We agree that our theoretical results hold without experimental validation because they are fully rigorous. However, we believe the experiments with additive Gaussian noise can help the community understand the results (as evidenced by the other reviews) and have thus decided to keep them.

---

> > ### Comment · Reviewer_sQPi · 2021-08-07
> > **Thanks for your response**
> >
> > Hi Author,
> >
> > Thanks for your feedbacks.
> >
> > I have also read the other reviews and your response to them. I will not change my rating. Although this paper is not about generalization error, it is still an important contribution to the optimization community: people usually inject noise to data when training, but there is little study to justify its correctness from the optimization aspect. There is some limitation on the theories, e.g. eta_t -> 0, but it is a good first step for this topic.

---

### Official Review · Reviewer_wBTQ · 2021-07-14

**Rating:** 6
**Confidence:** 2

**Summary:**

This paper studies the joint effect of data augmentation and learning rate scheduling on the optimization trajectory of overparameterized linear model. The authors study the conditions under which (S)GD will converge to the min-norm interpolator, even when the data augmentation distribution changes over time. With a more careful analysis, the authors also study the rate of convergence. Specific augmentations studied here are additive Gaussian noise and random projection.

**Limitations And Societal Impact:**

The limitations has been adequately addressed.

**Main Review:**

The proofs are technical and the overall problem studied is interesting. It is important to study how data augmentation affects the optimization trajectory. As the authors have mentioned, my main concerns are that 1) the model discussed here is limited to linear case, how can this case provide us insight for more general classes like kernels? and 2) although the problem formulation is general enough to include many types of augmentations, the detailed results are derived only for additive noise and random projects, which may not be what people mostly use nowadays.

What appears more concerning is that it is unclear to me why studying the convergence to the min-norm solution is a good idea. On a high level, augmenting with noise should make the learned minimizer more robust (at least to noise). As the authors have mentioned (and proven before), training with random noise corresponds to training with l2 regularization (eq 3.2). Therefore I'm not sure why the authors consider min-norm solution as the limiting classifier. I'm also curious what if the condition eta_t->0 is relaxed (at least is allowed to decay much slower), the limiting behavior may be more interesting.

**Time Spent Reviewing:**

3

---

> ### Author Response · Authors · 2021-08-05
> **Response to review**
>
> We thank the referee for their comments and suggestions, and respond to each below. We respectfully ask the referee to reconsider their initial evaluation on the basis of our responses.
>
> **Minimal norm solutions emerge as a natural inductive bias of our augmentations**
>
> The minimal norm solution occurs naturally in our setting as the inductive bias of augmented gradient descent (for additive noise and random projections) when the learning rate converges to 0. Though we agree it differs from the ridge regression optimum, we believe this illustrates that augmentation can have effects on optimization beyond its impacts on robustness.
>
> When \eta_t does not go to 0, our results also apply and show convergence to the ridge regression optimum. However, we chose not to consider this case since the objective is strongly convex and has a unique minimum.
>
> **Our techniques apply to kernel methods but with a new difficulty**
>
> Kernel methods may be written as y = W \phi(x) for an (infinite-dimensional) feature map \phi(x), so they are linear in \phi(x).  Thus, our framework applies after interpreting augmentations in feature space instead of input space.  If \phi is non-linear, the mapping between augmented data matrices X_t and their images \phi(X_t) may be complex, so interpreting our Theorems 3.1-3.2 for commonly used augmentations may not be straightforward.  This subtlety is why we chose to leave kernel methods to future work.
>
> **Additive noise and random projection as practical augmentations we can analyze in detail.**
>
> We chose additive Gaussian noise and random projections (as a proxy for CutOut) as simple models for augmentations used in practice (eg in computer vision).  Though other augmentations used in practice may be more complex, we wanted to illustrate concretely that augmented gradient descent has an implicit bias towards choosing minimal norm solutions.
>
> For more general augmentations, our main results Theorems 3.1 and 3.2 apply, but we have not specialized them to provide such an explicit characterization of the learned minimum beyond the general description in Theorem 3.1 and have therefore not discussed them in further detail.  However, for any specific augmentation used in practice, we may specialize our conditions (3.7)-(3.9).  As an example, we have briefly discussed how to place MixUp and label preserving transformations into our framework in lines 120-128.

---

> > ### Comment · Reviewer_wBTQ · 2021-09-02
> > **Concerns addressed**
> >
> > Thank the authors for their responses, my concerns are mostly addressed. Although I'm still not fully convinced that the min-norm interpolator is the best limiting estimator to look at (correct me if I'm wrong, but the min-norm interpolator only shows up in your particular coupling of data augmentation and learning rate decay. That is, a different learning rate decay can lead to a different limiting estimator), the fact that there exists a setting where the limiting estimator is the min-norm interpolator looks interesting enough to me. Hence I'm raising my score to 6.

---

> > > ### Author Response · Authors · 2021-09-03
> > > **Thanks for response**
> > >
> > > Thanks for this note.  We wanted to note in case it was not clear that our general Theorems 3.1 and 3.2 apply to other couplings of data augmentation and learning rate decay.  More specifically, the expressions in conditions (3.7) -- (3.12) are explicit functions of those quantities, and for any such schedule it is possible to specialize to results analogous to the ones we analyzed in detail in later sections.  In these cases, the limiting estimator is $W_\infty^* = \lim_{t \to \infty} W_t^*$ for $W_t^*$ defined in (3.4), which as you correctly note may differ from the min-norm interpolant depending on augmentation and learning rate choices.

---

### Official Review · Reviewer_cnWP · 2021-07-16

**Rating:** 6
**Confidence:** 3

**Summary:**

This paper rigorously studies the sufficient conditions of the convergence of GD/SGD when data augmentation is applied to learn an overparameterized linear model. Some of the sufficient conditions are also necessary. In particular, the sufficient conditions are formulated in terms of the learning rate, 2nd and 4th moment of augmented data matrices.

**Limitations And Societal Impact:**

Limitations and society impacts are adequately addressed.

**Main Review:**

-	This paper presents a solid theory that shed light on the understanding of the hyperparameters of data augmentation and GD/SGD, which affords many interesting insight, like Fig. 3.1. On the other hand, more numerical analysis with synthetic data is needed to further test the theory. In particular,
o	No numerical results are presented for the random projection augmentation in Sect. 5
o	It would be interesting to see if the sufficient conditions are sharp. If the selection of hyperparameters start to deviate from these conditions, would the error be slowly increasing, or would there be a phase transition from small error to large error?
-	Does the convergence result in Theorem 4.2 hold for any selection of minibatch size S_t that may be time dependent? Does the selection of the minibatch size and sigma_t need to jointly satisfy some constraints?
-	In Fig. 4.1, why does the MSE of Gauss exp schedule performs nearly the best for GD, but only outperforms the Gauss const for SGD? What is it in GD and SGD that leads to such difference?

**Time Spent Reviewing:**

4

---

> ### Author Response · Authors · 2021-08-05
> **Response to review**
>
> We thank the referee for their comments and suggestions, and respond to each below:
>
> **Our conditions are not necessary / sharp but already show unexpected interactions:**
>
> As we state at the end of Section 4.1, even for additive Gaussian noise our results are likely not sharp.  If we relax these conditions, as long as the learning rate converges to 0 and condition (3.7) holds, augmented GD will converge to the limit W_{\infty}^* of W_t^* defined in (3.4).   In this setting, if augmentation strength decays too fast so that (3.8) is violated, the optimum will still have small error, but not have minimal norm.  If augmentation strength decays too slowly so that (3.9) is violated, the norm of the optimum will be small, but the error will be large.  Our two conditions show that balancing these two considerations leads to the minimum norm solution.
>
> Thus, our conditions already show learning rate and augmentation scheme schedules combine in complex ways to influence the inductive bias of the learned minimizer.
>
> **Dependence of Theorem 4.2 on time-varying batchsize S_t:**
>
> This result holds for any time-dependent batch size with the same proof, since it is an asymptotic statement in t.  No constraint need hold on the relation between batch size and \sigma_t.
>
> **Experiments with random projections:**
>
> Our experimental results are meant to illustrate and sanity check our theorems (which we rigorously prove). Though we believe experiments on random projections would substantially duplicate insights from our existing results on additive noise, we ran similar experiments to those described in Section 4.3 for random projections with GD.  The results are similar to those shown in the left column of Figure 4.1, and we are happy to add them in the camera ready.
>
> **Gauss Exp in Fig 4.1:**
>
> Thanks for pointing this out; this was caused by an error in our implementation of sampling with replacement for SGD (this does not affect our experiments with GD).  We have corrected this error and regenerated the SGD plots.  The Unaugmented, Gauss const, and Gauss pow lines in the MSE plot for SGD do not exhibit qualitative changes from our Figure 4.1, while the Gauss exp line is extremely close to the Unaugmented line (as is the case for GD); the orthogonal weight norm plot is qualitatively unchanged.  Crucially, this does not affect any of our qualitative observations, and we will update this figure in the camera ready.

---

### Official Review · Reviewer_6v6C · 2021-07-16

**Rating:** 3
**Confidence:** 5

**Summary:**

This paper rigorously analyzed the convergence of data augmentation in linear regression under the convex setting. Its main result gives sufficient condition on the joint schedule of the learning rate and the augmentation strength under which asymptotic convergence rates are provided. Specializations The above result is then instantiated for additive Gaussian noise, random projections.

**Limitations And Societal Impact:**

No. See "Significance" in Main Review section.

**Main Review:**

Originality: The convergence results of data augmentation algorithms are solid. The analysis techniques seem non-trivial

Quality: Some assumptions seems to strong. How to justify the assumptions (3.10) (3.11) and (3.12) are reasonable? Something for assumptions (3.7) (3.8) and (3.9).

Clarity: Overall, the writing is clear to me.

Significance: First, I'm not convinced by the experimental results. It only runs on simulated data for additive Gaussian noise case. It would be better if the authors can provide the results for random projection case, and more importantly, if some real datasets can be used. I think additional experiments would have made the paper much stronger. Second, this paper only studies optimization of data augmentation. However, I think the most interesting and important thing is about the generalization of data augmentation, since the key goal of using data augmentation is to avoid over-fitting. Besides, it would also be better if the authors could provide some generalization results (such as testing error) in the experiments.

**Time Spent Reviewing:**

4

---

> ### Author Response · Authors · 2021-08-05
> **Response to review**
>
> We thank the referee for their comments and suggestions, and respond to each below. We respectfully ask the referee to reconsider their initial evaluation on the basis of our responses.
>
> **Our results demonstrate data augmentation has interesting impacts on optimization in addition to generalization.**
>
> Our paper falls into the vast line of work devoted to understanding the inductive bias of gradient-based optimization schemes. To our knowledge, prior to our work, there was no framework to study this inductive bias in settings with complex, realistic schedules for learning rate and augmentation scheme.
>
> Our results show that even for linear regression, these joint augmentation schedules affect the learned empirical risk minimizer in dramatic and complex ways and thus have substantial optimization impacts.  As such, we disagree that generalization is the only interesting thing to study about data augmentation and ask the referee to assess our paper in this context.
>
> **Conditions (3.7)-(3.12) in Theorems 3.1-2 are natural parallels of Monro-Robbins conditions from stochastic optimization.**
>
> We disagree these assumptions are too strong and believe them to be natural.  As we explain (abstract, contribution 1, and after Theorem 3.1), hypotheses (3.7)-(3.9) in Theorem 3.1 are analogs of classical Monro-Robbins conditions, which are necessary and sufficient for stochastic optimization of strongly convex objectives.  They naturally generalize these conditions “to situations where the stochastic optimization objective may change at each step” (and reduce to them for unchanging objectives).
>
> When specialized to additive noise (Theorem 4.1), we show these conditions hold for schedules with certain power law decays for learning rate and noise level, which are plausible in practice.
>
> **Experiments on real datasets / test error / random projections:**
>
> We theoretically study augmented gradient descent for linear regression as a simple setting where we can explicitly demonstrate the complex dependence of the inductive bias on parameter schedules.  Of course, practitioners do not generally train linear models using augmented gradient descent with random initialization, so we intend our experiments to illustrate and sanity check our theorems (which are rigorously proven) and not to make any direct applied claim.  As such, we are not sure what insights demonstrating our theorems on real datasets (augmented linear regression is not often used) or showing generalization results (generalization for linear models is determined entirely by the choice of optimum) would show.
>
> Though we believe experiments on random projections would substantially duplicate insights from our existing results on additive noise, we ran similar experiments to those described in Section 4.3 for random projections with GD.  The results are similar to those shown in the left column of Figure 4.1, and we are happy to add them in the camera ready.

---

### Decision · Program_Chairs · 2021-09-27

**Decision:**

Accept (Poster)

**Comment:**

As far as I am aware, this is the first work that presents theoretical results for optimization dynamics with data augmentation, and three of the four reviewers found the paper to be of sufficient significance and impact to warrant acceptance. I ultimately found the argument that the paper should be rejected on the grounds that it analyzes optimization dynamics instead of generalization error unconvincing. While I generally agree that the primary motivation for data augmentation is to prevent overfitting, it's not obvious to me that the effects of data augmentation on optimization are well understood or trivial, and it's not even a priori obvious to me that optimization should necessarily converge for all data augmentation schemes outside of basic ones like additive Gaussian noise.